# STEALTHY WORLD MODEL MANIPULATION VIA DATA POISONING

## ABSTRACT

Model-based learning agents that use a world model to predict and plan have shown impressive success in solving diverse, complex tasks and adapting to new environments. However, the process of exploring open environments and updating the model with collected experience also exposes them to adversarial manipulation. In this paper, we propose SWAAP, the first scalable and stealthy data poisoning method for world models, designed to benchmark their adversarial robustness. SWAAP uses a novel two-stage approach. In the first stage, the attacker identifies a target world model that deviates only slightly from the true environment but significantly degrades agent's performance when used for planning. This is achieved via a first-order bilevel optimization and a new transition gradient theorem. In the second stage, the attacker then performs the actual attack by perturbing a small subset of fine-tuning data to steer the fine-tuned world model toward the target model. Evaluations using diverse tasks show that our approach induces a substantial performance drop and remains effective even under robust training and detection, underscoring the urgent need for stronger protection in world modeling.

## 1 INTRODUCTION

While artificial intelligence (AI) has achieved remarkable success across various domains, building general-purpose agents that can quickly adapt to new tasks remains a major challenge, particularly for sequential decision-making tasks in open-ended environments that require substantial planning and adaptation. A promising direction is the development of *world models* (Ha & Schmidhuber, 2018) that accurately capture environmental structure and dynamics to support a wide range of downstream tasks. In this context, predictive world models, which allow agents to "imagine" future scenarios for safer, more efficient decisions, are proliferating (Hafner et al., 2023; Hansen et al., 2024; Zhou et al., 2025). Further, with recent advances in diffusion and transformer architectures, foundation world models (OpenAI, 2025; DeepMind, 2024; Nvidia, 2025a) capable of simulating interactive environments from multi-modal input are emerging and are increasingly being applied in complex domains such as autonomous vehicles and robotics (Nvidia, 2025b), making them valuable targets for malicious attacks.

To support effective decision-making across diverse domains, world models must encode broad knowledge, process high-dimensional inputs (e.g., images, videos, and text), and make long-horizon predictions, introducing new vulnerabilities not present in traditional supervised learning or model-free reinforcement learning (RL) systems. Despite extensive research in AI security and adversarial machine learning over the past decade, ensuring the robustness of world models against adversarial manipulation remains largely unexplored, limiting their deployment in high-stakes domains.

In this work, we take the first step toward adversarially robust world modeling by introducing poisoning attacks tailored to world models. Our method strategically alters trajectory data used for training or fine-tuning, with the objective of manipulating model-based decision-making while maintaining outputs close to those of a clean model to evade detection. We believe this line of work is both practical and influential, as it highlights a fundamental vulnerability in world models that underpins their reliability in downstream applications.

Traditional data poisoning methods from supervised learning (Biggio et al., 2012; Muñoz-González et al., 2017; Geiping et al., 2021) cannot be directly applied to our setting. These approaches typically assume fully differentiable training pipelines and discrete labels (e.g., flipping a *cat* to a *dog*),

allowing the adversary to optimize per-example perturbations via gradient-based techniques. In contrast, poisoning a world model requires identifying optimal perturbations of next states in transition tuples, which are continuous and affect long-horizon dynamics. Moreover, differentiating through the training process of the world model is computationally expensive and generally intractable, making standard supervised learning attacks unsuitable for this problem.

Existing data poisoning attacks in reinforcement learning (Rakhsha et al., 2020; Zhang et al., 2020) primarily manipulate the rewards or transitions observed during training to steer the learned policy toward an adversary-specified target policy. However, these approaches do not align with our setting for several reasons. First, they assume a fixed target policy, whereas in our case the adversary must instead identify a worst-case world model that maximizes performance degradation. Second, both (Rakhsha et al., 2020; Zhang et al., 2020) focus on relatively simple scenarios and rely on traditional convex optimization methods, which are computationally infeasible for large environments with continuous state and action spaces used in state-of-the-art world models such as TD-MPC2 (Hansen et al., 2024), DINO-WM (Zhou et al., 2025) and DreamerV3 (Hafner et al., 2023). These differences motivate the need for a fundamentally new approach to poisoning more complex world models.

In this work, we propose **SWAAP** (**S**tealthy **W**orld Model MA**nipulation via DA**ta **P**oisoning), a novel two-stage attack that manipulates the world model to mislead the agent and degrade its performance at test time. In the first stage, the attacker identifies a target poisoned world model that, once adopted by the agent, induces trajectories leading to low-return outcomes during testing. We formulate this as a bi-level optimization problem: the objective is to identify a poisoned world model that significantly degrades the agent's performance, subject to the constraint that the poisoned model must remain close to the original world model to avoid detection. In the second stage, we perform a data poisoning attack by injecting carefully crafted poisoned transitions into a newly collected fine-tuning dataset. This manipulation steers the updated world model toward the targeted poisoned model identified in the first stage. Instead of assuming an unrealistic model poisoning attack that directly replaces the original world model with the poisoned one, we adopt a more realistic data poisoning approach, where the attacker only manipulates the finetuning training data. This requires substantially less control over the system while still guiding the world model toward the targeted poisoned version. We evaluate SWAAP on three widely used continuous state-action environments: DMControl (Tassa et al., 2018), MyoSuite (Caggiano et al., 2022), and MetaWorld (Yu et al., 2020). By poisoning just 10% of a small fine-tuning dataset, our data poisoning attack can induce a significant drop in the agent's performance across diverse tasks. Moreover, the poisoned world models maintain a similar deviation from the true environment transitions, comparable to clean models, making them difficult to detect or mitigate. These findings highlight the urgent need for more robust world modeling techniques.

## 2 SYSTEM AND THREAT MODELS

In this section, we present the system and threat models, covering both world models and our proposed data poisoning attacks framework. A more detailed discussion of related work is provided in Appendix B.

### 2.1 WORLD MODELS

We study an agent that leverages a learned world model to interact with an environment formalized as a Markov decision process (MDP) $(S, A, P, R, \gamma)$, where $S$ is the state space, $A$ is the action space, $P : S \times A \to \Delta(S)$ is the transition kernel, $R : S \times A \to \mathbb{R}$ is the reward function, and $\gamma \in (0, 1)$ is the discount factor. During training, the agent learns a parameterized world model $P_\psi$ that approximates the environment dynamics: $P_\psi(s'|s, a) \approx P(s'|s, a)$. In practice, to improve computational efficiency and representation capacity, predictions are made in latent state $z = enc(s)$ with a learned encoder, and the model learns either through self-supervised learning on collected trajectories or by finetuning a pretrained foundation model (Zhou et al., 2025; Assran et al., 2025). To simplify the notation and keep the discussion general, we represent the world model as the transition probabilities between MDP states, $P_\psi(s'|s, a)$, in this and next sections. Our actual implementation in Section 4 operates in the latent space.

The fidelity of $P_\psi$ is critical, as it directly determines the effectiveness of downstream decision-making. The training or finetuning process $F(\psi_0, D_n)$ is carried out via stochastic gradient descent by minimizing the prediction error $\mathcal{L}(\psi_0; D_n) = \sum_{(s,a,s') \in D_n} \|s' - P_{\psi_0}(s,a)\|_2^2$. If the agent trains a world model from scratch, then $\psi_0$ denotes the randomly initialized model and $D_n = \{(s, a, s')\}$ represents a large training dataset. In contrast, if the agent relies on a pretrained world model $\psi_0$, which was trained on data that may slightly deviate from the true distribution encountered at test time or in environments with evolving dynamics, the model is typically updated intermittently with a smaller, newly collected dataset $D_n$. This continual updating from the pretrained model $\psi_0$ is regarded as finetuning.

Existing approaches differ in how the learned world model $P_\psi$ is used for action selection. For example, DINO-WM (Zhou et al., 2025) relies purely on model predictive control (MPC) without explicitly learning a policy, DreamerV3 (Hafner et al., 2023) follows a model-based reinforcement learning (MBRL) paradigm to train a policy directly from the world model, while TD-MPC2 (Hansen et al., 2024) combines both approaches by jointly training a policy and world model and using MPC for planning. These variations demonstrate that modern world models go beyond traditional notions of MBRL, as they are often motivated by building general-purpose agents capable of solving diverse tasks. In this work, we adopt the TD-MPC2 (Hansen et al., 2024) setting as our running example, while noting that our framework applies more broadly.

The agent's problem is to generate a policy $\pi_\theta(\psi)$ that maximizes the discounted cumulative return using the world model $P_\psi$ that approximate the true environment transition function $P$, which can be formulated as following,

$$\pi_\theta(\psi) = \arg\max_{\pi_\theta} J(P_\psi, \pi_\theta), \text{where } J(P_\psi, \pi_\theta) = \mathbb{E}_{P_\psi}\Big[\sum_{i=0}^{T} \gamma^i R(s_i, a_i)\Big]. \tag{1}$$

This formulation assumes the agent uses model-based reinforcement learning. If the agent employs model predictive control (MPC), then $J(P_\psi, \pi_\theta)$ also depends on the true environment dynamics $P$, which we omit from the notation for simplicity.

As discussed above, if the model predictive control (MPC) is used (Hansen et al., 2024; Zhou et al., 2025), the agent selects actions by searching over a finite-horizon sequence of candidate actions. Specifically, the planner first samples multiple action sequences. For each sampled action sequence $a_{t:t+H} = (a_t, a_{t+1}, \ldots, a_{t+H})$, the planner uses the world model $P_\psi$ to predict future states $\hat{s}_{t+i+1} \sim P_\psi(\cdot \mid \hat{s}_{t+i}, a_{t+i})$ and evaluates the expected discounted return of the corresponding trajectory. Then the agent finds the action sequence that can lead to the best expected discounted return and executes the first action $a_t$ and replans at the next time step $t + 1$, which is shown below.

$$\pi_\theta(s_t, \psi) = a_t^*, \text{where } a_{t:t+H}^* = \arg\max_{a_{t:t+H}} \mathbb{E}_{P_\psi} \sum_{i=0}^{H} \gamma^i R(s_{t+i}, a_{t+i}).$$

In contrast, DreamerV3 (Hafner et al., 2023) trains a policy $\pi_\theta(s)$ directly on imagined trajectories generated by $P_\psi$ using model-based reinforcement learning approach, allowing the agent to act without explicit planning at inference time. TD-MPC2 (Hansen et al., 2024) combines these approaches: it jointly trains a policy and world model, and then uses the policy as an initialization for MPC. We include the detailed MPC algorithm in the Appendix E for completeness.

## 2.2 DATA POISONING ATTACKS

Against the victim agent, the adversary seeks to degrade long-term return by corrupting the policy $\pi_\theta(\psi)$ through tampering with the world model update. We consider a data-poisoning adversary that may perturb a bounded fraction of transitions in the fine-tuning dataset $D_n$, replacing $(s, a, s')$ with $(s, a, \tilde{s}')$ to form a poisoned dataset $\tilde{D}_n$. We assume the adversary can observe the fine-tuning dataset $D_n$ and knows the world model architecture (white-box setting) but cannot directly overwrite the model or access the agent's policy parameters. The adversary is restricted to poisoning at most $r_p|D_n|$ samples, producing a modified dataset $\tilde{D}_n$. We denote $\|\tilde{D}_n - D_n\|_0$ as the number of transitions changed in $D_n$. We also assume the adversary can interact with a clean environment to collect transitions. Although we only consider poisoning the transition dynamics in this work,

our method can potentially apply to reward poisoning similar to (Rakhsha et al., 2020; Zhang et al., 2020). After the agent trains on $\tilde{D}_n$, the resulting world model $P_\psi$ deviates from the true dynamics $P$ and induces suboptimal behavior. This interaction can be formalized as the following bilevel optimization problem:

$$\min_{\tilde{D}_n} J\big(P, \pi_\theta(\psi)\big) + \lambda \mathrm{KL}(P_\psi \| P)$$

$$\text{s.t. } P_\psi = F(P_{\psi_0}, \tilde{D}_n), \pi_\theta(\psi) = \arg\max_{\pi_{\theta'}} J(P_\psi, \pi_{\theta'}), \|\tilde{D}_n - D_n\|_0 \le r_p |D_n|, \tag{2}$$

where $F(P_\psi, D_n)$ denotes the fine-tuning process of the world model on dataset $D_n$, given a prior world model $P_\psi$, $\mathrm{KL}(P_\psi \| P)$ is the average KL divergence over state-action pairs that constrains the poisoned world model to remain close to the true transition dynamics, thereby evading detection, and $\lambda$ is a parameter that balances the two objectives. The *outer minimization* corresponds to the adversary's objective of reducing real-environment performance, while the *inner maximization* captures the agent's policy optimization under the poisoned world model.

## 3 STEALTHY WORLD MODEL MANIPULATION VIA DATA POISONING

Directly solving the bilevel optimization in Eq. 2 is highly intractable. Because it requires differentiating through the finetuning process of the world model, which is typically non-transparent and computationally expensive. In addition, unlike standard data poisoning in supervised learning (e.g., flipping labels from *cat* to *dog*) Biggio et al. (2012); Muñoz-González et al. (2017), our setting involves finding optimal perturbations of next states in transition tuples, a substantially more complex problem. As a result, traditional gradient-based data poisoning methods cannot be directly applied. Existing RL data poisoning approaches (Zhang et al., 2020; Rakhsha et al., 2020) also do not transfer

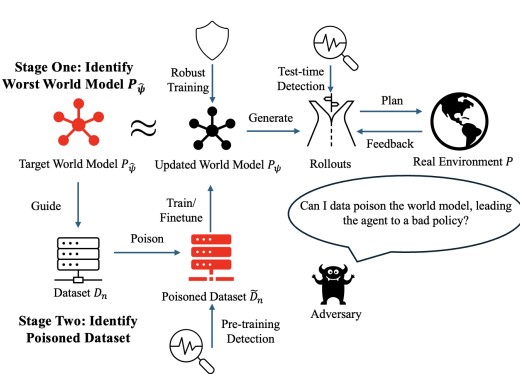

Figure 1: Pipeline of SWAAP.

to our setting due to: (i) they require a pre-specified target policy, (ii) their methods cannot scale to complex continuous state and action spaces. To address these challenges, our method SWAAP decomposes the adversary's optimization into a two-stage procedure (see Figure 1).

In the first stage, we identify a worst-case target world model $\hat{\psi}$ by formulating a bilevel optimization problem of a two-player game between the agent that maximizes the return by using the model and the attacker that minimizes the return by changing the model (Eq. 3). In the second stage, we design poisoned transitions within $D_n$ through the gradient matching technique (Geiping et al., 2021) to craft $\tilde{D}_n$ such that the gradient calculated during the finetuning procedure will steer the world model toward $\hat{\psi}$. This decomposition avoids direct differentiation through the training process while providing a principled way to construct adversarial data to provide a surrogate optimization problem to the original bilevel data poisoning optimization problem.

To ensure our attack remains stealthy, it must avoid detection at both the data-poisoning and testing stage. During poisoning, injected transitions $(s, a, \tilde{s}')$ should not be distinguishable from genuine data; during testing, the learned world model $P_\psi$ should remain close to the environment's true transition function $P$. We enforce stealthiness by introducing two regularization terms—one applied at the data level to constrain per-sample perturbations, and one applied at the model level to limit deviations of the learned dynamics, balancing attack effectiveness with detectability.

### 3.1 STAGE 1: IDENTIFICATION OF PERTURBED MODELS

In the first stage, the attacker formulates a model poisoning problem, seeking a perturbed model state $\psi$ such that when the agent uses this model for planning under the true environment dynamics $P$, its expected return is minimized.

As discussed in Section 2.1, the agent may leverage the world model in different ways when deriving its policy, and the specifics of the actual online planning algorithm it uses may be hidden from the attacker. To keep our approach general, we consider a surrogate policy $\pi_\theta(\psi)$ that mimics a model-based RL agent, where $\pi_\theta(\psi)$ is derived to maximize the return in the imaginary environment $P_\psi : \pi_\theta(\psi) \in \arg\max_{\pi_{\theta'}} J(P_\psi, \pi_{\theta'})$.

The adversary thus minimizes the return of the surrogate policy that learns from perturbed transitions by solving the following bilevel optimization problem:

$$\begin{aligned} \min_\psi \quad & J(P, \pi_\theta(\psi)) + \lambda \mathrm{KL}(P_\psi \| P) \\ \text{s.t.} \quad & \pi_\theta(\psi) \in \arg\max_{\pi_{\theta'}} J(P_\psi, \pi_{\theta'}). \end{aligned} \tag{3}$$

The goal is to minimize the KL-regularized expected return with respect to the optimal policy derived from the poisoned environment.

This bilevel optimization problem remains challenging due to its nested structure. The primary difficulty lies in computing the derivative $\nabla_\psi \pi_\theta(\psi)$. If one directly treats $\pi_\theta(\psi)$ as a function of $\psi$, gradient-based methods such as hypergradient descent (Bard, 2013) update $\psi$ using

$$\nabla_\psi J(P, \pi_\theta(\psi)) = \nabla_\psi \pi_\theta(\psi) \nabla_2 J(P, \pi_\theta(\psi)),$$

but evaluating $\nabla_\psi \pi_\theta(\psi) = \nabla_\psi \arg\max_{\pi_{\theta'}} J(P_\psi, \pi_{\theta'})$ requires solving linear systems and computing costly second-order Hessians. Prior work has attempted to bypass this difficulty either through surrogate approximations or heuristics that are difficult to control (Pedregosa, 2016; Ghadimi & Wang, 2018; MacKay et al., 2019). Moreover, reinforcement learning objectives are highly non-convex, involve high-dimensional state–action spaces, and present a complex optimization landscape. As a result, naive bilevel optimization methods often fail to converge reliably, making them unsuitable for Problem 3 (Liu et al., 2021).

To obtain a more scalable solution, we adopt the first-order dynamic barrier gradient descent method in (Liu et al., 2022), which reformulates the problem by replacing the implicit argmin operator with a value-function constraint. This allows the outer objective and inner objective to be solved without explicitly computing the derivative $\nabla_\psi \pi_\theta(\psi)$. Formally, let $f(\psi, \theta) := J(P, \pi_\theta) + \mathrm{KL}(P_\psi \| P)$ denote the outer objective, and $q(\psi, \theta) := \max_{\pi_{\theta'}} J(P_\psi, \pi_{\theta'}) - J(P_\psi, \pi_\theta)$, which measures the suboptimality of $\pi_\theta$ relative to the optimal policy under $P_\psi$. Under the value-function approach, the bilevel problem becomes the following constrained optimization:

$$\min_{\psi, \theta} f(\psi, \theta) \quad \text{s.t.} \quad q(\psi, \theta) \leq 0.$$

The main idea of the dynamic barrier gradient method is to iteratively update the parameters $(\psi, \theta)$ to reduce $f$ while ensuring that $q$ decreases whenever $q > 0$. To this end, the parameters $(\psi, \theta)$ are jointly updated as:

$$(\psi_{k+1}, \theta_{k+1}) \leftarrow (\psi_k, \theta_k) - \xi \left[ \nabla f(\psi_k, \theta_k) + \lambda_k \nabla q(\psi_k, \theta_k) \right], \tag{4}$$

where $\xi$ is the step size and $\lambda_k$ is the dual variable at iteration $k$ (see Appendix D for details). The gradients of $f$ and $q$ can be explicitly written as:

$$\nabla_{(\psi, \theta)} f(\psi_k, \theta_k) = \left( \lambda \nabla_\psi \mathrm{KL}(P_{\psi_k} \| P), \nabla_\theta J(P, \theta_k) \right), \tag{5}$$

$$\nabla_{(\psi, \theta)} q(\psi_k, \theta_k) \approx \left( \nabla_\psi J(\psi_k, \theta_k^W) - \nabla_\psi J(\psi_k, \theta_k), -\nabla_\theta J(\psi_k, \theta_k) \right), \tag{6}$$

where we use the shorthand $J(\psi_k, \theta_k) := J(P_{\psi_k}, \pi_{\theta_k})$, and $\theta_k^W$ is the $W$-step approximation of the optimal policy $\pi_\theta^* \in \arg\max_{\pi_{\theta'}} J(P_\psi, \pi_{\theta'})$.

While the gradient of the expected return with respect to the policy $\theta$ is readily available from the classic policy gradient theorem (Sutton et al., 1999), the gradient with respect to the transition $\psi$ is not directly available. To address this, we establish the following result by extending the policy gradient theorem (see Appendix C for the proof).

**Theorem 1.** *The transition gradient of expected return in an MDP with transition dynamics $P_\psi$, policy $\pi_\theta$, and the state-value function $V(s) := \mathbb{E}_{P_\psi, \pi_\theta} \left[ \sum_{i=0} \gamma^i R(s_{t+i}, a_{t+i}) \Big| s_t = s \right]$ can be expressed by*

$$\nabla_\psi J(P_\psi, \pi_\theta) = \mathbb{E}_{(s, a, r, s') \sim P_\psi, \pi_\theta} \left[ (R(s, a, s') + V(s')) \nabla_\psi \log P_\psi(s' | s, a) \right].$$

---

**Algorithm 1** Algorithm for Perturbed Model Identification

---

Input: Environment dynamics $P$, a dynamics model $\psi_0$, a policy $\theta_0$
Output: perturbed model $\hat{\psi}$
**while** $(\psi_k, \theta_k)$ not converged **do**
    $\theta_k^0 \leftarrow \theta_k$
    **for** $i \leftarrow 1$ to $W$ **do**
        $\theta_k^{i+1} \leftarrow \theta_k^i + \nabla_\theta J(\psi_k, \theta_k^i)$
    Collect rollouts from $(\psi_k, \theta_k^W), (P, \theta_k), (\psi_k, \theta_k)$ to form $\text{buffer}_{kW}, \text{buffer}_{pk}, \text{buffer}_{kk}$,
    respectively
    $\Omega \leftarrow [\,]$
    **for** $i \leftarrow 1$ to $N$ **do**
        Compute $\nabla_\psi J(\psi_k, \theta_k^W)$ by sampling $\text{buffer}_{kW}$
        Compute $\nabla_\theta J(P, \theta_k)$ by sampling $\text{buffer}_{pk}$
        Compute $\nabla_\psi J(\psi_k, \theta_k)$ and $\nabla_\theta J(\psi_k, \theta_k)$ by sampling $\text{buffer}_{kk}$
        Compose $\nabla f(\psi_k, \theta_k)$ and $\nabla q(\psi_k, \theta_k)$ from Equation 5 and Equation 6, and $\lambda_k$
        Append $\nabla f(\psi_k, \theta_k) + \lambda_k \nabla q(\psi_k, \theta_k)$ to $\Omega$
    $\omega_k \leftarrow \frac{\sum_{\omega \in \Omega} \omega}{N}$
    $(\psi_{k+1}, \theta_{k+1}) \leftarrow (\psi_k, \theta_k) - \xi \cdot \omega_k$
**return** $\hat{\psi} \leftarrow \psi_k$

---

Using this formulation, our algorithm for perturbed model identification proceeds as follows. We initialize $(\psi_0, \theta_0)$ from pretrained models and iteratively approximate the locally optimal policy via $T$-step gradient ascent. At each iteration, we collect rollouts under $P_{\psi_k}, \pi_{\theta_k}$ and compute the required gradients of the return with respect to $\psi$ and $\theta$. The parameters are then updated using Equation 4. To improve sample efficiency, we maintain replay buffers to reuse trajectories, enabling multiple gradient updates per step. The full procedure is summarized in Algorithm 1.

## 3.2 STAGE 2: POISONING DATA TO MANIPULATE MODEL

As noted earlier, it is unrealistic for an attacker to directly replace the agent's world model $P_\psi$ with $P_{\hat{\psi}}$, as this would require complete control over the victim agent. Thus, we choose to conduct a data poisoning attack on the fine-tuning dataset $D_n$ to manipulate the victim's model toward $P_{\hat{\psi}}$, which can be stated as follows.

$$\min_{\tilde{D}_n} \ \mathrm{KL}\big(P_{\hat{\psi}} \| F(P_{\psi_0}, \tilde{D}_n)\big) \quad \text{s.t.} \ \|\tilde{D}_n - D_n\|_0 \leq r_p |D_n|, \tag{7}$$

where $P_\psi = F(P_{\psi_0} \| \tilde{D}_n)$ is the poisoned world model after fine-tuning and $\mathrm{KL}\big(P_{\hat{\psi}} \| F(\psi_0, \tilde{D}_n)\big)$ is the average KL divergence over state-action pairs between the target world model generated in Stage 1 and the updated world model using poisoned data. To improve stealth, the attacker is allowed to modify up to $r_p |D_n|$ transitions in the finetuning dataset $D_n$, where $r_p$ is the maximum fraction of transitions that can be perturbed.

Directly solving (7) requires jointly optimizing which transitions to modify and how to modify them, which is computationally hard: selecting up to $r_p |D_n|$ transitions from $D_n$ is a discrete subset selection problem, and for each such choice, one must also solve a continuous perturbation optimization over the chosen elements. Solving these two problems jointly requires searching an exponentially large space while repeatedly finetuning (or differentiating through) the world model, rendering the approach computationally infeasible in practice.

We address the challenge of jointly determining which transitions to modify and how to modify them by adopting a simple yet effective heuristic. Specifically, we first select the top $r_p$ fraction of transitions $\{(s, a, s')\} \subset D_n$ that exhibit the largest model residuals, where for a given dynamics model $\psi$ the residual is defined as

$$e_\psi(s, a, s') = \|s' - P_\psi(s, a)\|_2. \tag{8}$$

Let $D_p$ denote the selected top $r_p$ fraction of transitions with residuals with respect to the target model $e_{\hat{\psi}}(s, a, s')$ in $D_n$, and let $D_c = D_n \setminus D_p$ be the remaining clean transitions. The poisoned

dataset is then $\tilde{D}_n = D_c \cup D_p$. We then solve the reduced data poisoning problem only in $D_p$ using gradient matching (Geiping et al., 2021) as follows. Let

$$G_{\text{real}} = \mathbb{E}_{(s,a,\tilde{s}') \in \tilde{D}_n}\left[\nabla_{\psi_0}\|\tilde{s}' - P_{\psi_0}(s,a)\|_2^2\right], G_{\text{target}} = \mathbb{E}_{(s,a,s') \sim D_{\text{all}}}\left[\nabla_{\psi_0}\|P_{\hat{\psi}}(s,a) - P_{\psi_0}(s,a)\|_2^2\right],$$

here $D_{\text{all}} = \{(s,a,s')\}$ is a large dataset collected by the attacker. The gradient matching problem is

$$\min_{\{\tilde{s}_i' : (s_i,a_i,s_i') \in D_p\}} \left\|G_{\text{real}} - G_{\text{target}}\right\|_2^2 \quad \text{s.t.} \quad \tilde{D}_n = D_c \cup D_p, |D_p| \leq r_p|D_n|. \tag{9}$$

This gradient-matching formulation provides an effective solution to the original data poisoning problem in Equation 7 because it directly aligns the updates induced by the poisoned dataset $\tilde{D}_n$ with the target world model $P_{\hat{\psi}}$. By defining $G_{\text{real}}$ as the expected gradient on the poisoned data and $G_{\text{target}}$ as the gradient corresponding to the adversary's worst-case model over a large reference dataset $D_{\text{all}}$, minimizing $\|G_{\text{real}} - G_{\text{target}}\|_2^2$ ensures that training on $\tilde{D}_n$ nudges the world model $P_\psi$ in the direction toward the target world model $P_{\hat{\psi}}$.

To solve this gradient matching problem, we define a loss function $L_P$ with two components:

$$L_P = (1-\alpha)\big(1 - \cos(G_{\text{real}}, G_{\text{target}})\big) + \alpha \sum_{(s_i,a_i,s_i') \in D_p} |\|\tilde{s}_i' - s_i'\|_2 - \|P_\psi(s_i,a_i) - s_i'\|_2|. \tag{10}$$

The first term maximizes the cosine similarity between the gradient from the poisoned dataset and the target gradient estimated from $D_{\text{all}}$, thereby steering the fine-tuned world model toward $P_{\hat{\psi}}$. The second term focuses on the selected subset $D_p$ and leverages the fact that the original world model $P_\psi$ already exhibits some one-step prediction error (see Table 1). Thus, we can measure perturbation size relative to this existing error to allow bounded, plausible poisoned data while discouraging large or conspicuous changes to avoid being detected. Regularization coefficient $\alpha \in (0,1)$ controls the trade-off between gradient alignment and the relative size of perturbations compared to the model's inherent prediction error. By minimizing the loss $L_P$, the adversary determines the poisoned transitions $D_p$ and forms the poisoned finetuned dataset $\tilde{D}_n$.

### 3.3 DEFENSES AGAINST DATA POISONING

We assume that the agent may employ defense mechanisms during finetuning to mitigate the effects of data poisoning. We consider pre-training detection, robust training, and test-time detection defenses. Pre-training detection and robust training methods require the agent to have a relatively accurate mimic model of the environment transitions; however, in practice, the agent may not possess such a model due to limited clean data. To overestimate the agent's defense capability in our experiments, we assume that the agent has access to a reasonably accurate world model $P_{\psi'}$ to conduct these defenses.

Detection-based methods identify potentially poisoned transitions before they are used for model updates by comparing the residual against a certain threshold. Transitions with residuals $e_{\psi'}(s,a,s')$ that violate this threshold are flagged as suspicious and removed or down-weighted (Chen et al., 2021).

Training-time defenses aim to reduce the impact of poisoned samples without explicitly identifying them. One representative approach is the TRIM strategy (Biggio et al., 2012), which iteratively filters transitions based on their residuals. At each iteration, the transitions are ranked by $e_{\psi'}(s,a,s')$, and only the lowest $(1-\beta)n$ residual transitions are retained, where $n$ is the number of transitions considered and $\beta \in (0,1)$ controls the fraction of discarded data. The finetuning update is then computed using this subset. By discarding high-residual transitions, the TRIM strategy limits the influence of adversarially perturbed transitions while preserving the underlying clean dynamics.

In addition, the agent can perform a test-time detection to monitor whether the world model is poisoned. The agent can observe the true next state $s'$ from the environment and compare it with the world model output $P_\psi(s,a)$. We define $\delta = \frac{1}{T}\sum_{i=0}^{T} e_\psi(s_i,a_i,s_i') = \frac{1}{T}\sum_{i=0}^{T}\|s' - P_\psi(s,a)\|_2$ as the deviation, where $T$ is the number of transitions $(s,a,s')$ the agent observes during testing.

## 4 EXPERIMENTS

### 4.1 EXPERIMENT SETUP

We adopt TD-MPC2 (Hansen et al., 2024) as our victim algorithm because it jointly trains a policy and a world model, then uses the learned policy to initialize MPC. We evaluate a finetuning scenario that the agent starts from a pretrained world model $\psi_0$, which is trained from $1,000,000$ clean transitions, and updates it using a small dataset $D_n$ including $5,000$ transitions. TD-MPC2 uses 512 rollouts with a horizon of 3 in MPC planning. A detailed hyperparameter table is in Appendix F.

We report the cumulative return and the deviation $\delta$ from the true transitions under four scenarios. The **Clean** corresponds to the agent using the pre-trained, unpoisoned world model $P_\psi$ during testing. **SWAAP (Random)** uses randomly perturbed transitions as the target model in stage two instead of the poisoned world model $P_{\hat{\psi}}$ from stage one. **Direct Model Poisoning** deploys the target poisoned world model $P_{\hat{\psi}}$ without data poisoning, representing an unrealistic direct model overwrite. Finally, **SWAAP** reports the results from the two-stage attack pipeline, including both target model identification and data poisoning. We conduct our experiments on three widely used benchmarks with continuous state and action spaces: DMControl (Tassa et al., 2018), MyoSuite (Caggiano et al., 2022), and MetaWorld (Yu et al., 2020). Additional results for SWAAP, SWAAP(Random), direct model poisoning, and ablation on MPC parameters are provided in Appendix G.

### 4.2 ATTACK PERFORMANCES

Table 1 shows our SWAAP algorithm can significantly lower the agent's return while maintaining a comparable level of deviation as a clean world model. SWAAP (Random) is substantially less effective than our two-stage attack at comparable deviation levels $\delta$, which underscores the value of Stage one for identifying a target world model. Figure 2 further illustrates that allowing larger deviations (by relaxing the regularization coefficient $\alpha$) yields greater return degradation in Humanoid-Walk and Myo-Pen-Twirl; additional ablations on $\alpha$ appear in Appendix G.1.

Relative to the direct model poisoning results, performance depends on the magnitude of the deviation: in some environments, direct model overwrite produces stronger immediate degradation, while in others our two-stage pipeline attains comparable or better results. This indicates two points: (i) realistic data poisoning with constrained budget and stealthiness can closely approximate the effect of direct model poisoning, and (ii) in environments such as Myo-Pen-Twirl-Hard, our Stage two procedure does not fully recover the Stage one target $P_{\hat{\psi}}$, suggesting room for improved data-poisoning algorithms. To visualize these relationships, we plot return versus deviation curves for varying $r_p$ and $\alpha$ across the three methods (SWAAP, SWAAP(Random), and direct model poisoning) in Figure 3. In Humanoid-Walk, our method achieves the strongest attack for a given $\delta$, while in some Myosuite tasks, direct model poisoning remains the most damaging under the same level of $\delta$.

The higher variance of SWAAP shown in Table 1 is primarily caused by the reward structure of these environments rather than by our attack method itself. In tasks such as pen-twirl-hard, a successful attack episode drives the agent into a failure trajectory and produces extremely low returns (e.g., around 1,000), whereas an unsuccessful attack episode still behaves similarly to clean execution and achieves high returns (e.g., around 5,000). Even clean policies naturally exhibit noticeable variance in these environments because occasional failures lead to large return drops. Under attack, this effect becomes more pronounced: the mixture of low-reward failure episodes and high-reward normal episodes produces a wider spread of returns.

### 4.3 ATTACK UNDER DEFENSES

We consider three defenses the agent might deploy: pre-training detection (Chen et al., 2021), robust training (TRIM) (Biggio et al., 2012), and testing-time detection. As illustrated in Fig. 4a, the residuals of the poisoned dataset $\tilde{D}_n$ are statistically indistinguishable from those of a clean dataset $D_n$, indicating the agent cannot reliably identify poisoned transitions during pre-training. Fig. 4b reports results when the agent applies the TRIM robust-training procedure: our SWAAP attack preserves its effectiveness under TRIM and in some cases produces even stronger degradation. We conjecture this occurs because the poisoned transitions are carefully crafted to mislead TRIM's filtering rule into

Table 1: Main results comparison between clean, direct model poisoning, SWAAP (Random), and SWAAP. Each test aggregates over 100 episodes and all have the same data poisoning ratio $r_p = 0.1$, $\alpha$ is chosen to be 0.9 to make $\delta$ small and comparable to clean ones.

| Environment | | Clean | | SWAAP (Random) | | Direct Model Poisoning | | SWAAP | |
|---|---|---|---|---|---|---|---|---|---|
| | | Return | $\delta$ | Return | $\delta$ | Return | $\delta$ | Return | $\delta$ |
| DMControl | humanoid-walk | $866 \pm 57$ | $.09 \pm .05$ | $839 \pm 90$ | $.10 \pm .05$ | $813 \pm 112$ | $.11 \pm .05$ | $521 \pm 332$ | $.14 \pm .08$ |
| | humanoid-run | $587 \pm 53$ | $.05 \pm .03$ | $565 \pm 60$ | $.05 \pm .04$ | $207 \pm 42$ | $.10 \pm .04$ | $497 \pm 64$ | $.06 \pm .04$ |
| | dog-walk | $744 \pm 49$ | $.06 \pm .02$ | $803 \pm 36$ | $.06 \pm .02$ | $328 \pm 76$ | $.10 \pm .03$ | $693 \pm 75$ | $.06 \pm .03$ |
| | dog-run | $653 \pm 54$ | $.05 \pm .02$ | $631 \pm 54$ | $.05 \pm .02$ | $246 \pm 66$ | $.08 \pm .02$ | $475 \pm 63$ | $.06 \pm .02$ |
| | cheetah-run | $834 \pm 62$ | $.03 \pm .04$ | $845 \pm 34$ | $.02 \pm .04$ | $415 \pm 121$ | $.09 \pm .09$ | $792 \pm 42$ | $.04 \pm .06$ |
| Myosuite | pen-twirl-hard | $3693 \pm 2195$ | $.06 \pm .03$ | $2507 \pm 2343$ | $.13 \pm .10$ | $2876 \pm 2364$ | $.07 \pm .02$ | $2479 \pm 2236$ | $.13 \pm .13$ |
| | reach-hard | $733 \pm 98$ | $.04 \pm .04$ | $631 \pm 510$ | $.07 \pm .05$ | $678 \pm 162$ | $.07 \pm .02$ | $531 \pm 772$ | $.10 \pm .13$ |
| Metaworld | push | $1789 \pm 20$ | $.08 \pm .03$ | $1747 \pm 185$ | $.09 \pm .06$ | $1572 \pm 559$ | $.11 \pm .06$ | $1716 \pm 58$ | $.11 \pm .07$ |
| | soccer | $1707 \pm 43$ | $.05 \pm .04$ | $1691 \pm 180$ | $.04 \pm .05$ | $1642 \pm 89$ | $.07 \pm .06$ | $1483 \pm 516$ | $.06 \pm .08$ |

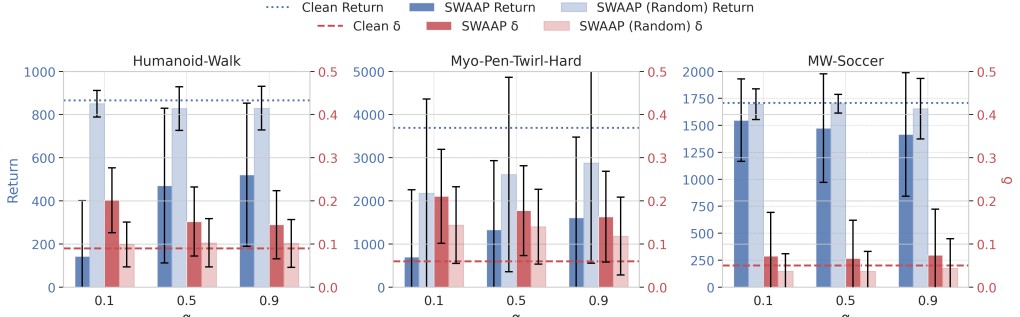

Figure 2: Comparing results between SWAAP (Random) and SWAAP across different $\alpha = (0.1, 0.5, 0.9)$. Data poisoning ratio $r_p$ used by the three tasks are $0.1, 0.2, 0.1$, respectively. It shows, with some increase on deviation, our attack can reliably lead to decrease of agent's performance, and the failure of SWAAP (Random), which brew the poison data by applying random noise for the model to finetune, to affect the agent in humanoid-walk and mw-soccer indicate that the identification of vulnerable perturbed model state is positively contributing to attack result.

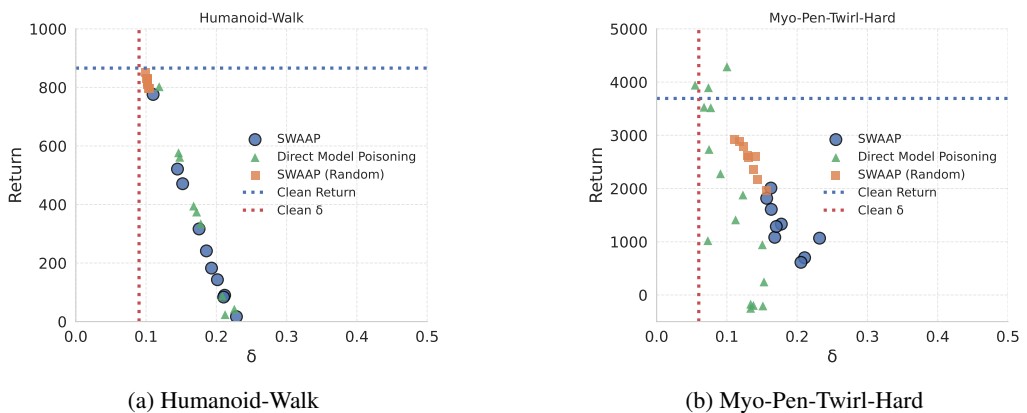

(a) Humanoid-Walk        (b) Myo-Pen-Twirl-Hard

Figure 3: Comparing SWAAP (Random), direct model poisoning, and SWAAP in $\delta$–Return plots. The data points for SWAAP (Random) and SWAAP are from using different $r_p$ and $\alpha$, while data points for direct model poisoning baseline are extracted from different training iterations under $\lambda = 10$ for humanoid-walk, $\lambda = 1$ for myo-pen-twirl-hard (see Figure 7).

removing clean transitions, thereby amplifying the attack's effect after training. Finally, as shown in Table 1, SWAAP induces a level of model deviation at test time that is comparable to—or indistinguishable from—that produced by a clean world model, demonstrating that the attack remains difficult to detect during testing.

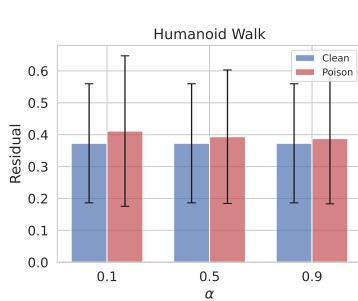

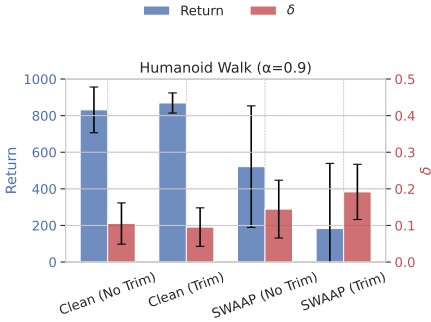

(a) Humanoid-Walk Pre-Training Detection

(b) Humanoid-Walk TRIM Training Defense

Figure 4: Detection and training defense results. $\beta = 0.2$ is used in TRIM training in (b).

## 5 CONCLUSION AND FUTURE WORK

In this work, we proposed **SWAAP**, a novel two-stage data poisoning attack that manipulates world models, leading to significant performance degradation at test time while remaining stealthy. A promising future direction is to further refine the data poisoning stage by designing algorithms that align the poisoned model more closely with the target model obtained from the first stage. Another interesting avenue is to study foundation world models that can be applied across diverse tasks, and to investigate how their generality influences both the effectiveness of poisoning attacks and the robustness of potential defenses.

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

APPENDIX

# A    USE OF LLMS

In this work, we use LLMs mainly for grammar checking and rephrasing words. All references were independently verified by the authors. No algorithms, proofs, or experimental results were generated by ChatGPT, and no proprietary or sensitive data were shared with the tool. All technical contributions and analyses are solely the authors' work.

# B    RELATED WORKS

## B.1    WORLD MODELS

World models aim to learn compact and predictive representations of environment dynamics that can be used for planning and control. Instead of interacting directly with the environment, an agent can rely on its learned model to simulate trajectories, evaluate policies, and anticipate future states. First introduced in (Ha & Schmidhuber, 2018), this work introduced a three-component architecture combining a variational autoencoder (VAE), a recurrent neural network (RNN), and a linear controller, demonstrating that policies trained entirely within a latent model can transfer back to the real environment. This work sparked a line of research exploring increasingly powerful and scalable model-based reinforcement learning frameworks.

Dreamer-style agents use latent dynamics models optimized end-to-end with reinforcement learning. DreamerV3 (Hafner et al., 2023) achieves state-of-the-art performance on visual control and robotic tasks. It employs a recurrent state-space model (RSSM) with deterministic state $h_t$ and stochastic latent $s_t$, performing rollouts entirely in latent space. Three heads are trained on the RSSM: a reward predictor $\hat{r}_t$, a value function $\hat{V}(\cdot)$, and an actor $\pi$, using imagined rollouts. At decision time, the actor proposes candidate actions, either sampled or taken as the mean.

More recent approaches have investigated architectural advances tailored for large-scale and complex environments. DINO-WM (Zhou et al., 2025) leverages self-supervised vision transformers (ViTs) to improve perception quality, enabling stronger generalization in visually rich settings. Similarly, TD-MPC2 (Hansen et al., 2024) proposes a temporally abstracted model predictive control framework that combines world models with trajectory optimization, achieving sample-efficient learning and strong performance in high-dimensional continuous control tasks.

Parallel to these developments, diffusion-based world models have emerged as a promising alternative (Pearce et al., 2023; Janner et al., 2022). By parameterizing the transition distribution as a denoising diffusion process, these models can capture multi-modal and stochastic dynamics more effectively than conventional Gaussian latent models. Diffusion world models have been shown to improve both planning quality and robustness to uncertainty, making them attractive in settings where dynamics are highly non-deterministic.

## B.2    MODEL-BASED REINFORCEMENT LEARNING

Model-based reinforcement learning (MBRL) is a powerful paradigm that improves sample efficiency and enables better generalization compared to purely model-free methods. In MBRL, an agent learns an explicit dynamics model of the environment and leverages this model for planning or policy optimization. Formally, we consider a Markov decision process (MDP) $(S, A, P, R, \gamma)$, where $S$ is the state space, $A$ is the action space, $P : S \times A \to \Delta(S)$ defines the transition kernel, $R : S \times A \to \mathbb{R}$ is the reward function, and $\gamma \in (0, 1)$ is the discount factor. The goal is to find a policy $\pi : S \to \Delta(A)$ that maximizes the expected discounted return $\sum_{i=0}^{T} \gamma^i R(s_t, a_t)$.

Unlike model-free approaches that learn value functions or policies directly from experience (Mnih et al., 2015; Lillicrap et al., 2016; Haarnoja et al., 2018), MBRL explicitly learns a parametric approximation $\hat{P}_\psi$ of the transition kernel $P$, often using neural networks (Deisenroth & Rasmussen, 2011; Chua et al., 2018; Janner et al., 2019; Hansen et al., 2024). This learned dynamics model can then be used for model-predictive control (MPC) (Camacho & Bordons, 2013), trajectory opti-

mization (Tassa et al., 2012), or to generate synthetic rollouts for policy improvement (Kaiser et al., 2020; Hafner et al., 2019).

### B.3 MODEL POISONING ATTACK

Model poisoning attacks directly compromise a learned model by altering its parameters. Model poisoning attack assumes the adversary can craft malicious model updates to steer the learning outcome or directly replace the model parameters. For example, carefully crafted local malicious updates can harm the performance of a federated learning system (Fang et al., 2020).

In reinforcement learning, model poisoning is particularly concerning when applied to the agent's learned dynamics model. Small, adversarial modifications to the transition kernel can propagate over planning horizons, misleading policy improvement and causing degraded performance (Rakhsha et al., 2020). This makes model poisoning a uniquely severe threat in model-based RL, as even subtle deviations from the true dynamics can cascade into large errors in long-term decision making.

### B.4 DATA POISONING ATTACKS AND DEFENSES

Data poisoning attacks compromise learning systems by corrupting the training dataset to bias the learned model toward an adversarial objective. Unlike model poisoning, which manipulates parameters directly, data poisoning assumes the attacker can only influence the data stream but not the learning algorithm itself, which is a more realistic attack scenario. Data poisoning attacks have been shown to significantly degrade the model performance even with small amounts of poisoned data (Biggio et al., 2012). A recent and effective data poisoning technique leverages gradient matching, which optimizes poisoned samples such that their gradients closely align with those of a target adversarial objective (Geiping et al., 2021). By ensuring that poisoned data induces updates similar to those of the adversary's intended solution, gradient matching enables stealthy and highly effective poisoning even under limited attacker control. We also adopt gradient matching in the second stage of our attack methods, where we injects carefully crafted poisoned transitions into the newly collected fine-tuning dataset.

In reinforcement learning, data poisoning is particularly dangerous since training relies on sequential interactions with the environment. By injecting corrupted transitions into the replay buffer or modifying observed trajectories, an adversary can degrade the long-term performance of the agent, or even embed targeted failures (Rakhsha et al., 2020; Zhang et al., 2020).

Effective defenses against data poisoning attacks involve detection and training-time strategies. We consider two widely used data poisoning defenses in our work. Chen et al. (2021) propose De-Pois, an attack-agnostic detection method that identifies poisoned data points with the help of a mimic model trained from clean data samples. By measuring the difference between the samples on the mimic model's outputs, De-Pois can flag and remove suspicious points, improving robustness without assuming knowledge of the attack type.

Complementing detection, training-time defenses aim to mitigate the effect of poisoned samples during learning. For example, Biggio et al. (2012) introduce the TRIM method and its iterative variant, which estimate and remove a fraction of potentially poisoned points based on residual errors and statistical properties of the data.

## C DERIVATION OF TRANSITION GRADIENT

**Theorem 1.** *The transition gradient of expected return in an MDP with transition dynamics $P_\psi$ and policy $\pi_\theta$ can be expressed by*

$$\nabla_\psi J(P_\psi, \pi_\theta) = \mathbb{E}_{(s,a,r,s')\sim P_\psi, \pi_\theta}\Big[(R(s,a,s') + V(s'))\nabla_\psi \log P_\psi(s'|s,a)\Big].$$

**Proof:** To prove the theorem, we start from the derivative of the state-value function (approximated by a differentiable neural network) of an arbitrary initial state $s_0$ and extend the state into the future time indefinitely, and expand it to write it as a recursive form in terms of $M$ function defined below, from which it unrolls to become a infinite series of the sum of $M(s_k)$ weighted by the probability

of reaching $s_k$ from $s_0$ in $k$ steps.

$$\nabla_\psi V(s_0) = \nabla_\psi \sum_{a_0} \pi(a_0|s_0) Q(s_0, a_0)$$

$$= \nabla_\psi \sum_{a_0} \pi(a_0|s_0) \sum_{s_1} P_\psi(s_1|s_0, a_0)[R(s_0, a_0, s_1) + V(s_1)]$$

$$= \sum_{a_0} \pi(a_0|s_0) \sum_{s_1} \nabla_\psi P_\psi(s_1|s_0, a_0)[R(s_0, a_0, s_1) + V(s_1)]$$

$$+ \sum_{a_0} \pi(a_0|s_0) \sum_{s_1} P_\psi(s_1|s_0, a_0) \nabla_\psi V(s_1)$$

$$= M(s_0) + \sum_{s_1} \rho_\pi(s_0 \to s_1, k=1) \nabla_\psi V(s_1) \qquad \text{a recursive relation about } \nabla_\psi V(s_i)$$

$$= M(s_0) + \sum_{s_1} \rho_\pi(s_0 \to s_1, 1)\Big[M(s_1) + \sum_{s_2} \rho_\pi(s_1 \to s_2, 1)\nabla_\psi V(s_2)\Big]$$

$$= M(s_0) + \sum_{s_1} \rho_\pi(s_0 \to s_1, 1)M(s_1) + \sum_{s_2} \rho_\pi(s_0 \to s_2, 2)M(s_2) + \cdots \qquad \text{unrolling into future steps indefinitely}$$

$$= \sum_x \sum_{k=0}^\infty \rho_\pi(s_0 \to x, k)M(x) \qquad \text{swap order of the summation and re-indexing}$$

$$= \sum_s \eta(s)M(s) \qquad \text{let } \eta(s) = \sum_{k=0}^\infty \rho_\pi(s_0 \to s, k)$$

$$\propto \sum_s \frac{\eta(s)}{\sum_s \eta(s)}M(s) \qquad \frac{\eta(s)}{\sum_s \eta(s)} \text{ is the stationary distribution of } s$$

$$= \sum_s d_\pi(s) \sum_a \pi(a|s) \sum_{s'} \nabla_\psi P_\psi(s'|s, a)(R(s, a, s') + V(s')).$$

where $M(s_0) := \sum_{a_0} \pi(a_0|s_0) \sum_{s_1} \nabla_\psi P_\psi(s_1|s_0, a_0)[R(s_0, a_0, s_1) + V(s_1)]$, and the scaling from dividing the normalization constant $\sum_s \eta(s)$ can be absorbed into the learning rate.

$\rho_\pi(s_0 \to s_1, k = 1) := \sum_a \pi(a|s)P(s_1|s_0, a_0)$ is the transition probability of reaching $s_1$ from $s_0$ at step $k = 1$, and the relation $\rho^\pi(s \to x, k + 1) = \sum_{s'} \rho^\pi(s \to s', k)\rho^\pi(s' \to x, 1)$ is used to write the transition gradient into recurrence form. Therefore,

$$\nabla_\psi J(P_\psi, \pi) = \sum_s d_\pi(s) \sum_a \pi(a|s) \sum_{s'} \nabla_\psi P_\psi(s'|s, a)(R(s, a, s') + V(s'))$$

$$= \mathbb{E}_{(s,a,r,s')\sim P_\psi, \pi}[(R(s, a, s') + V(s'))\nabla_\psi \log P_\psi(s'|s, a)].$$

For deterministic transitions $s' = \mu_\psi(s, a)$, this becomes

$$\nabla_\psi J(P_\psi, \pi) = \nabla_\psi \sum_s d_\pi(s)V(s)$$

$$= \nabla_\psi \sum_s d_\pi(s) \sum_a \pi(a|s)Q(s, a)$$

$$= \nabla_\psi \sum_s d_\pi(s) \sum_a \pi(a|s) \sum_{s'} P_\psi(s'|s, a)[R(s, a, s') + V(s')]$$

$$= \nabla_\psi \sum_s d_\pi(s) \sum_a \pi(a|s)[R(s, a, \mu_\psi(s, a)) + V(\mu_\psi(s, a))]$$

$$= \sum_s d_\pi(s) \sum_a \pi(a|s)\nabla_{s'}[R(s, a, s') + V(s')]|_{s'=\mu_\psi(s,a)}\nabla_\psi \mu_\psi(s, a).$$

## D  MODEL POISONING ALGORITHM

To study adversarial robustness of world models, we cast model poisoning as a bilevel optimization problem where the outer objective aims to degrade the return $J(P, \pi_\theta)$ under the true environment dynamics $P$, while the inner objective enforces that the perturbed model $P_\psi$ still admits a locally optimal policy. Directly differentiating through the inner optimization $\nabla_\psi \pi_\theta(\psi)$ is intractable due to the implicit dependence of $\pi_\theta$ on $\psi$ and the need for costly Hessian computations.

Following the first-order dynamic barrier gradient descent method, we reformulate the problem as a constrained optimization that balances the decrease of the outer objective $f(\psi, \theta)$ and a constraint function $q(\psi, \theta)$.

$$\begin{aligned} \min_\psi \quad & J(P, \pi_\theta) + \lambda L(P_\psi, P) \\ \text{s.t.} \quad & \min_{\pi'_\theta} J(P_\psi, \pi'_\theta) - J(P_\psi, \pi_\theta) \leq 0 \end{aligned} \tag{11}$$

and define $f(\psi, \theta) := J(P, \pi_\theta) + L(P_\psi, P_0)$ and $q(\psi, \theta) := J(P_\psi, \pi_\theta^*) - J(P_\psi, \pi_\theta) = \min_{\pi'_\theta} J(P_\psi, \pi'_\theta) - J(P_\psi, \pi_\theta)$. This transformed problem is solvable by iteratively update $(\psi, \theta)$ to decrease $f$ while at the same time keeping the constraint $q \leq 0$ satisfied by decreasing $q$ whenever $q > 0$ in each step:

$$(\psi_{k+1}, \theta_{k+1}) \leftarrow (\psi_k, \theta_k) - \xi \omega_k \tag{12}$$

$$\text{where} \quad \omega_k = \arg\min_\delta ||\nabla f(\psi_k, \theta_k) - \omega||^2 \tag{13}$$

$$\text{s.t.} \ \langle \nabla q(\psi_k, \theta_k), \omega \rangle \geq \phi_k \tag{14}$$

this could be solved in closed form, which gives $\omega_k = \nabla f(\psi_k, \theta_k) + \lambda_k \nabla q(\psi_k, \theta_k)$, with $\lambda_k = \max\left(\frac{\phi_k - \langle \nabla f(\psi_k, \theta_k), \nabla q(\psi_k, \theta_k) \rangle}{\|\nabla q(\psi_k, \theta_k)\|^2}, 0\right)$ and $\phi_k$ is chosen to be $\eta q(\psi, \theta)$ or $\eta \|\nabla q(\psi, \theta)\|^2$. Therefore, the procedure to optimize $f$ by jointly updating $(\psi, \theta)$ is:

$$(\psi_{k+1}, \theta_{k+1}) \leftarrow (\psi_k, \theta_k) - \xi[\nabla f(\psi_k, \theta_k) + \lambda_k \nabla q(\psi_k, \theta_k)] \tag{15}$$

where $\nabla f(\psi_k, \theta_k) = \nabla_{(\psi_k, \theta_k)} f(\psi_k, \theta_k)$ is the gradient update of outer problem and $\nabla q(\psi, \theta) = \nabla_{(\psi, \theta)} q(\psi, \theta)$ imposes the constraint. Expressed explicitly, the gradient of $f$ and $q$ are:

$$\nabla_{(\psi, \theta)} f(\psi_k, \theta_k) = (\lambda \nabla_\psi L(\psi_k, P), \nabla_\theta J(P, \theta_k)) \tag{16}$$

$$\nabla_{(\psi, \theta)} q(\psi_k, \theta_k) \approx \left(\nabla_\psi J(\psi_k, \theta_k^T) - \nabla_\psi J(\psi_k, \theta_k), -\nabla_\theta J(\psi_k, \theta_k)\right) \tag{17}$$

where the shorthand $J(\psi_k, \theta_k) = J(P_{\psi_k}, \pi_{\theta_k})$ is used, and $\theta_k^T = \pi_{\theta_k}^T$ is the $T$ steps approximation of $\pi_{\theta_k}^* \in \arg\max_{\pi'_\theta} J(P_\psi, \pi'_\theta)$.

## E  MODEL PREDICTIVE CONTROL

---

**Algorithm 2** General Model Predictive Control (MPC)

---

1: **Input:** World model $\psi$, current state $s_t$, goal $o_g$ (optional), horizon $H$, number of rollouts `num_samples`
2: Encode: $z_t = \text{enc}(s_t)$, $z_g = \text{enc}(o_g)$ if goal is given
3: **for** each MPC step **do**
4:     Sample `num_samples` candidate action sequences $\{a_{0:H-1}^i\}_{i=1}^{\texttt{num\_samples}}$ (from a policy prior, an action sampler, or learned actor)
5:     **for** each sequence $i$ **do**
6:         Roll out latent trajectory $\hat{z}_{1:H}^i = P_\psi(\hat{z}_t, a_{0:H-1}^i)$
7:         Evaluate cost or return:

$$J^i = \begin{cases} \sum_{h=0}^{H-1} \gamma^h r(\hat{z}_h^i, a_h^i) + \gamma^H Q(\hat{z}_H^i) & \text{(value/bootstrap)} \\ \|\hat{z}_H^i - z_g\|^2 & \text{(goal-closeness)} \\ \text{other task-specific objective} \end{cases}$$

8:     Select best sequence(s) according to $J^i$
9:     Refit a sampling distribution to top-$K$ sequences
10: **return** first action or first $k$ actions from selected sequence or sampler

---

We provide a general formulation of Model Predictive Control (MPC), which abstracts across variants such as TD-MPC2 (Hansen et al., 2024), DINO-WM (Zhou et al., 2025). In this section, we occasionally use $\hat{z}$ to denote imagined latent, compare to real latent $z$ encoded from real state $s$ from environment.

To model the dynamics, TD-MPC2 (Hansen et al., 2024) learns the world model $P_\psi(z'|z, a)$ that predicts next latent state given current latent state and action, where $z = enc(s)$ is the latent state from a learned encoder, and its agent plans by using the learned world model $P_\psi$ together with a learned prior policy $\pi_{\text{prior}}$. At each planning step the planner first obtains `num_pi_trajs` candidate trajectories by rolling out $\pi_{\text{prior}}$ under transition $P_\psi$; these yield the policy-guided action sequences. The planner then samples an additional `num_samples − num_pi_trajs` random action sequences from a stochastic proposal (e.g., i.i.d. or Gaussian-perturbed sequences). During the inner improvement/refit loop the action components coming from the prior policy are treated as guidance and are typically held fixed, while only the remaining (random) action sequences are optimized. This hybrid design reduces search dimensionality and biases search toward plausible behavior while still allowing exploratory refinement.

DINO-WM (Zhou et al., 2025) does not use a learned prior. Instead it draws many random action sequences, rolls each sequence forward in the world model $P_\psi$ for horizon $H$, and scores each rollout by a final-state objective (the distance to a goal latent $z_g$, e.g., $J = \|\hat{z}_H - z_g\|_2$). The top-performing sequences are retained as elites and the sampling distribution is refit to those elites (CEM-style); this process repeats until the sampling distribution concentrates on sequences that reach the goal.

The above MPC algorithm highlights the shared structure: rolling out candidate action sequences over a planning horizon $H$ using a learned world model $P_\psi$, evaluating them under a task-specific objective (e.g., bootstrapped return, goal closeness, or other criteria), and executing the first action of the best sequence. The number of sampled rollouts `num_samples` and the planning horizon $H$ are hyperparameters that directly affect the quality of planning and computation cost. The general MPC procedure is summarized in Algorithm 2.

## F  Hyperparameters

We report our SWAAP key hyperparameters in Table 2. Other parameters are the same as stated in TD-MPC2 (Hansen et al., 2024).

## G  More Results

### G.1  SWAAP and SWAAP (Random)

We show more results on SWAAP and SWAAP(Random) in Figure 5, comparing the return and $\delta$ of various tasks under SWAAP, SWAAP(Random) with the clean return and clean $\delta$ at different poison ratio $r_p$. A higher $\alpha$ constrains model deviation more strictly, reducing attack strength but improving stealth. A lower $\alpha$ relaxes the deviation constraint and increases performance degradation. A higher data poison rate $r_p$ will also strengthen our attack performance.

Empirically, across all tested $\alpha$ and $r_p$, we did not observe gradient explosion or vanishing. Optimization remained numerically stable throughout Stage 2.

### G.2  Pre-Training Detection

We show more results on pre-training detection results in Figure 6, showing the residual of the data before and after poisoning at different data poisoning regularization coefficient $\alpha$.

### G.3  Model Poisoning Training Curve

We show the stage one model training curve in Figure 7, which comes from testing the agent that directly uses the perturbed model for 10 episodes after every few iteration of updates.

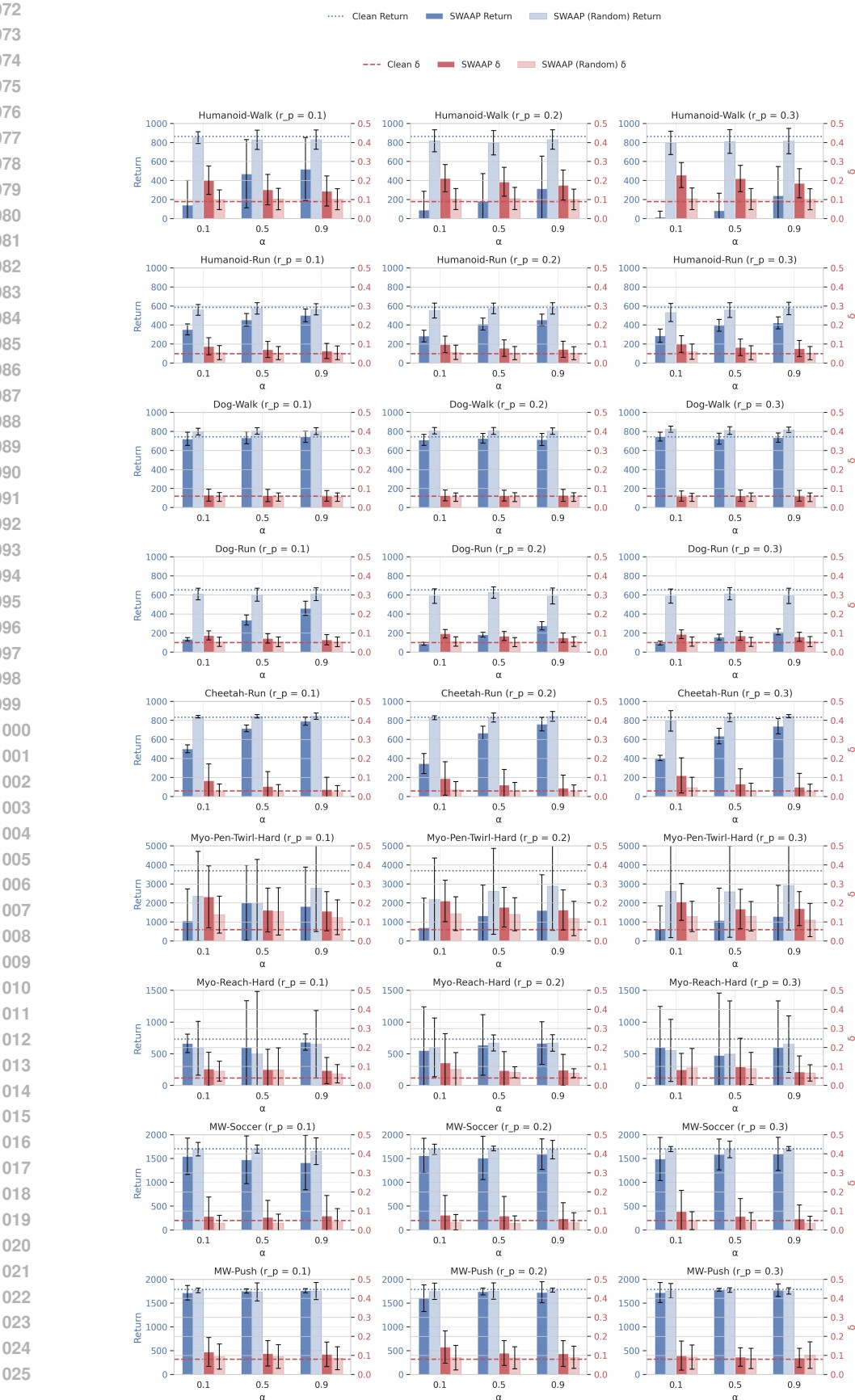

Figure 5: SWAAP and SWAAP (Random) more results

Table 2: Key hyperparameters used in experiments; other parameters are the same as in TD-MPC2

| Hyperparameter | Typical value(s) | Description |
|---|---|---|
| **Model Poisoning Hyperparameters** | | |
| obs_dim | 17-223 | The range of observation dimensions for tasks |
| action_dim | 6-39 | The range of action dimensions for tasks |
| latent_dim | 512 | The latent state dimension to encode observation state |
| H | 3 | Planning horizon length, which is the number of rollout steps into the future when planning |
| num_samples | 512 | Total number of trajectories for planning, including policy trajectories and random trajectories |
| policy_samples | 24 | number of trajectories for planning that are from policy prior |
| iterations | 6 | number of iterations done to optimize the actions in planning |
| $\lambda$ | $\{0, 1, 10, 100\}$ | Consistency coefficient applied to the $L(P, P_\psi)$ term during model poisoning in Equation 3 |
| $W$ | $\{30\}$ | The number of update steps used to approximate $\theta_k^*$ |
| $N$ | $\{16\}$ | The size of $\Omega$, the number of small updates aggregated to compute gradients of $f_k, q_k$ and $\lambda_k$ |
| sample_batch_size | $\{256\}$ | Number of transitions used to calculate each sample in $\Omega$ |
| num_step | $\{100, 500\}$ | The imagined rollout length, 500 for DMControl tasks and 100 for others. |
| buffer_size | $\{500\}$ | Buffer size for every buffer used in Algorithm 1. |
| **Data Poisoning Hyperparameters** | | |
| $r_p$ | $\{0.1, 0.2, 0.3\}$ | Fraction of the dataset (or proportion of trajectories/transitions) modified by adversary. |
| $\alpha$ | $\{0.1, 0.5, 0.9, 0.95, 0.99\}$ | Data poisoning regularization coefficient in Equation 10 |
| poison_steps | $\{5000\}$ | The number of gradient matching update steps. |
| noise_scale | $\{0.1, 0.3, 0.5\}$ | The random perturbation scale of SWAAP (Random) |
| train_epochs | $\{100, 500\}$ | The number of training epochs for finetuning, 100 for DMControl tasks and 500 for others. |
| learning_rate | $\{0.01, 0.0001\}$ | Learning rate of finetuning, 0.01 for MyoSuite and 0.0001 for others. |
| $|D_n|$ | $\{5000\}$ | Size of the finetuning dataset. |
| $|D_{\text{all}}|$ | $\{50000\}$ | Size of the dataset to approximate $G_{\text{target}}$ |

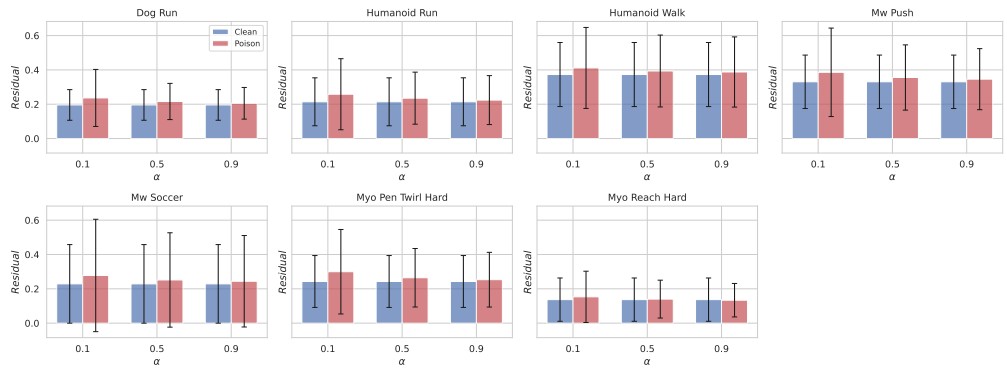

Figure 6: More Pre-Training detection results

## G.4 MPC ABLATION

Table 3 reports the reward of TD-MPC2 under different rollout horizons ($h = 3, 6, 9$) and number of samples for both the clean and SWAAP settings. We observe that our SWAAP attack consistently

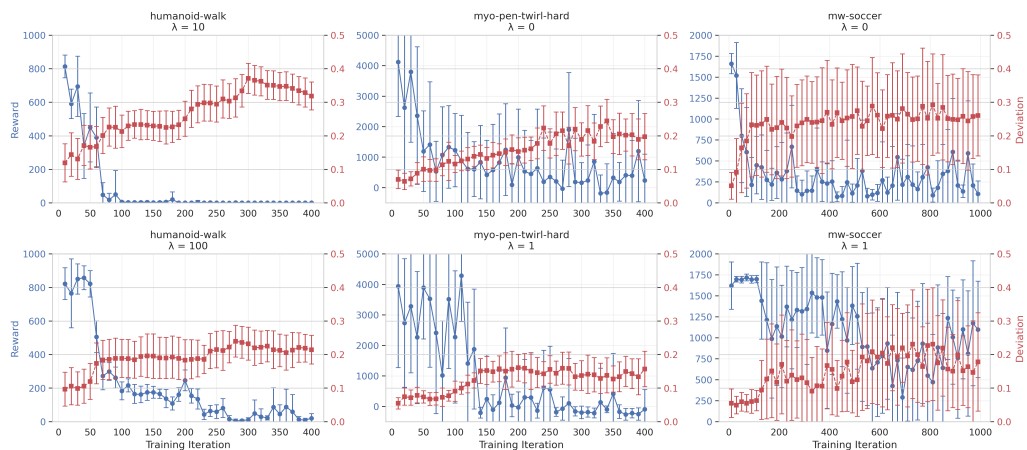

Figure 7: Model poisoning curves (referred to as Direct Model Poisoning in Table 1) of return and $\delta$ tested across different training iteration under different $\lambda$, each testing is average of 10 test episodes. From these plots, it is seen that $\lambda$ is suppressing the deviation of model, but the varying influence of $\lambda$ over different environment implies that different $\lambda$ is required to constrain the perturbation of model.

Table 3: Effect of MPC hyperparameters (`num_samples`, horizon $H$) on reward under clean and SWAAP settings. Values are mean $\pm$ std over 10 episodes.

| Num Samples | | Return | | |
|---|---|---|---|---|
| | | $h = 3$ | $h = 6$ | $h = 9$ |
| 64 | Clean | $12 \pm 21$ | $5 \pm 4$ | $9 \pm 21$ |
| | SWAAP | $7 \pm 7$ | $19 \pm 39$ | $14 \pm 20$ |
| 128 | Clean | $150 \pm 95$ | $58 \pm 92$ | $2 \pm 2$ |
| | SWAAP | $101 \pm 85$ | $38 \pm 51$ | $43 \pm 41$ |
| 256 | Clean | $788 \pm 157$ | $488 \pm 279$ | $68 \pm 115$ |
| | SWAAP | $347 \pm 257$ | $93 \pm 93$ | $21 \pm 36$ |
| 512 | Clean | $855 \pm 62$ | $687 \pm 112$ | $37 \pm 61$ |
| | SWAAP | $594 \pm 203$ | $49 \pm 61$ | $41 \pm 58$ |
| 1024 | Clean | $875 \pm 63$ | $812 \pm 91$ | $385 \pm 287$ |
| | SWAAP | $656 \pm 192$ | $87 \pm 147$ | $75 \pm 92$ |

Table 4: Attack performance with different perturb ratio in Stage 1 in Humanoid-Walk with $\alpha = 0.9$

| Stage 1 poison fraction | Return | $\delta$ |
|---|---|---|
| 0% (clean) | $866 \pm 57$ | $0.09 \pm 0.05$ |
| 10% | $825 \pm 106$ | $0.10 \pm 0.06$ |
| 50% | $669 \pm 242$ | $0.12 \pm 0.07$ |
| 100% | $521 \pm 332$ | $0.14 \pm 0.08$ |

reduces the agent's performance across all configurations, demonstrating its effectiveness under every setting. Additionally, the MPC performance generally increases with the number of samples and decreases as the rollout horizon grows. Consequently, the impact of our attack is also influenced by these hyperparameters: it tends to be more significant when the baseline MPC performance is higher (larger sample sizes) and slightly less effective at longer horizons given the low clean reward.

### G.5 ABLATION ON FRACTION OF POISONED STATE-ACTION PAIRS

Stage 1 of SWAAP can be adjusted to only perturb a certain fraction of the state-action pairs' transitions. We have conducted additional experiments on perturbing the top 10% and 50% most influential state-action pairs compared to perturbing all pairs considered before in Humanoid-Walk with

Table 5: Attack performance with different data poison ratio ($r_p$) in Stage 2 in Humanoid-Walk with $\alpha = 0.9$

| $r_p$ | Return | $\delta$ |
|---|---|---|
| clean | $866 \pm 57$ | $0.09 \pm 0.05$ |
| 0.01 | $844 \pm 94$ | $0.10 \pm 0.05$ |
| 0.05 | $739 \pm 172$ | $0.11 \pm 0.07$ |
| 0.10 | $521 \pm 332$ | $0.14 \pm 0.08$ |
| 0.20 | $317 \pm 339$ | $0.18 \pm 0.08$ |
| 0.30 | $242 \pm 305$ | $0.19 \pm 0.08$ |

Table 6: Additional evaluation on MyoSuite and MetaWorld environments.

| Environment | Natural Return | SWAAP Return | Natural $\delta$ | SWAAP $\delta$ |
|---|---|---|---|---|
| myo-obj-hold-hard | $-289 \pm 2343$ | $-2671 \pm 4241$ | $0.10 \pm 0.07$ | $0.13 \pm 0.05$ |
| myo-pose | $696 \pm 4$ | $665 \pm 157$ | $0.04 \pm 0.04$ | $0.08 \pm 0.06$ |
| myo-key-turn | $1125 \pm 196$ | $940 \pm 375$ | $0.19 \pm 0.14$ | $0.11 \pm 0.08$ |
| mw-coffee-pull | $1511 \pm 25$ | $1402 \pm 318$ | $0.08 \pm 0.04$ | $0.09 \pm 0.06$ |
| mw-door-open | $1551 \pm 58$ | $1310 \pm 445$ | $0.04 \pm 0.02$ | $0.06 \pm 0.05$ |

$r_p = 0.1, \alpha = 0.9$ in Table 4. The result shows that SWAAP achieves a tunable balance between attack effectiveness and stealthiness, allowing the attacker to induce small or moderate return drops that are within the range of typical world-model deviation.

### G.6 ABLATION ON $r_p$

We conduct an ablation study on different $r_p$ ratios in Stage 2, with results in Table 5. We observe that with lower poisoning rates (1% and 5%), the attack performance decreases as expected, but it still consistently induces noticeable degradation in return.

### G.7 COMPUTATIONAL COST

Stage 1 takes approximately 1 hour on DMControl and Metaworld and 2 hours on Myosuite. Stage 2 takes roughly 10 minutes. Our attack designs intentionally reduce computational overhead: (1) Stage 1 uses a first-order dynamic barrier method, avoiding the expensive Hessian computations typically required in bilevel optimization; (2) Stage 2 employs a modified gradient-matching step that selects the most influential data points for poisoning instead of exhaustively searching over all possible combinations, which significantly reduces computational cost.

### G.8 RESULTS FOR ADDITIONAL ENVIRONMENTS

We conducted additional experiments on five MyoSuite and MetaWorld tasks, with results reported in Table 6. All experiments use $r_p = 0.1$ and are evaluated over 100 episodes. For `Obj-Hold-Hard`, `Key-Turn`, and `Coffee-Pull`, we set $\alpha = 0.9$, while for `Pose` and `Door-Open`, we use $\alpha = 0.1$. These results further confirm that SWAAP reliably reduces the return while keeping the poisoned world model close to the clean model, consistent with our main findings.

### G.9 ABLATION ON FINE-TUNING DATASET SIZE

We conducted additional experiments on `HumanoidWalk` using fine-tuning dataset sizes ranging from 5,000 to 25,000, with $\alpha = 0.9$ and $r_c = 0.1$. The results are shown in Table 7. Although increasing the fine-tuning dataset size reduces the magnitude of performance degradation, as larger clean buffers naturally dilute the poisoned portion, SWAAP still consistently induces meaningful return drops across all tested sizes, demonstrating its robustness even when the victim fine-tunes on substantially larger datasets.

Table 7: Effect of fine-tuning dataset size in HumanoidWalk.

| Fine-tuning Dataset Size | Return | $\delta$ |
|---|---|---|
| 5,000 | $521 \pm 332$ | $0.14 \pm 0.08$ |
| 10,000 | $540 \pm 348$ | $0.14 \pm 0.08$ |
| 15,000 | $550 \pm 306$ | $0.14 \pm 0.08$ |
| 20,000 | $607 \pm 300$ | $0.13 \pm 0.08$ |
| 25,000 | $679 \pm 234$ | $0.12 \pm 0.07$ |

Table 8: Key hyperparameters used in the DINO-WM Push-T experiment

| Hyperparameter | Typical value(s) | Description |
|---|---|---|
| obs_shape | (3, 224, 224) | Shape of pixel observation $s$. |
| latent_shape | (196, 394) | Shape of latent state $z = \text{enc}(s)$. |
| action_dim | 10 | Dimension of actions. |
| num_hist | 3 | Number of historical latent states used to predict the next state. |
| H | 5 | Planning horizon length (number of rollout steps during planning), which also corresponds to the number of planned actions. |
| num_samples | 100 | Total number of sampled trajectories during planning. |
| opt_steps | 30 | Number of optimization iterations per planning step using cross-entropy method (CEM). |

Table 9: Gray-box attack on TD-MPC2 multi-task world model

| Source Task | Target Task | Setting | Return (R $\pm$ std) | $\delta$ ($\pm$ std) |
|---|---|---|---|---|
| MW-Soccer | MW-Soccer | Natural | $1380 \pm 382$ | $0.11 \pm 0.11$ |
| MW-Soccer | MW-Soccer | SWAAP | $1115 \pm 442$ | $0.13 \pm 0.12$ |
| MW-Soccer | MW-Push | Natural | $1290 \pm 609$ | $0.13 \pm 0.16$ |
| MW-Soccer | MW-Push | SWAAP | $1064 \pm 692$ | $0.16 \pm 0.19$ |

### G.10 DINO-WM RESULTS

We adapted our method to the DINO-WM world models for goal-conditioned tasks (Zhou et al., 2025). Our experimental setup largely follows the configuration used in the original implementation; the detailed settings are provided in Table 8.

For Stage 1, we used a goal-conditioned RL agent as the surrogate policy. Preliminary results show that the identified target world model reduces the success rate on the Push-T task from 92% to 72%, with the relative deviation increasing from $\delta = 0.11 \pm 0.06$ to $\delta = 0.24 \pm 0.10$.

We then conducted additional experiments incorporating Stage 2 of our attack pipeline. Using the perturbed model identified in Stage 1, we employed gradient matching to generate poisoned samples (poisoning ratio 0.15) and fine-tuned a DINO-WM model on the resulting poisoned dataset. On the Push-T task, performance dropped from 92% to 77%, with $\delta = 0.16 \pm 0.09$. These early results on DINO-WM and goal-conditioned tasks demonstrate that our method extends naturally to more complex and diverse settings.

### G.11 GRAY-BOX ATTACK ON TD-MPC2 MULTI-TASK WORLD MODEL

We conducted a preliminary gray-box experiment in which the attacker only has access to a surrogate world model trained on a single task, while the victim employs a multi-task world model. The attacker performs Stage 1 and Stage 2 entirely on the surrogate model and then supplies the resulting poisoned dataset to the victim for fine-tuning. In this experiment, the surrogate is a world model trained on MW-Soccer, whereas the victim is a multi-task world model used across multiple tasks. We use $r_p = 0.1, \alpha = 0.1$ in this experiment. Our initial results show that SWAAP still induces substantial performance degradation across multiple downstream tasks (see Table 9), despite the mismatch in between the surrogate and victim models.

