# OpenReview forum: "Stealthy World Model Manipulation via Data Poisoning"
_ICLR.cc/2026/Conference — Submitted to ICLR 2026_

### Official Review · Reviewer_K1NY · 2025-10-20

**Soundness:** 3
**Presentation:** 3
**Contribution:** 3
**Rating:** 4
**Confidence:** 4

**Summary:**

This paper introduces SWAAP, the first scalable, stealthy data poisoning attack targeting world models in model-based reinforcement learning (MBRL) agents. World models (e.g., DreamerV3, TD-MPC2, DINO-WM) enable agents to simulate environments for planning and adaptation in complex domains like robotics and autonomous driving. However, their reliance on fine-tuning with collected trajectories exposes them to adversarial manipulation. SWAAP degrades agent performance by subtly altering fine-tuning data, while keeping the poisoned model close to clean dynamics to evade detection. Evaluations across DMControl, MyoSuite, and MetaWorld show substantial return drops (e.g., up to 50% in some tasks) with just 10% poisoned data, persisting under defenses like TRIM and detection methods.

**Strengths:**

- SWAAP's decomposition of the bilevel poisoning problem into target model identification and gradient-matching data poisoning is elegant and practical. It avoids intractable direct differentiation during training, making it scalable to complex, high-dimensional environments—unlike prior RL attacks, which are limited to simple or policy-focused scenarios.

- Addresses emerging risks in high-stakes domains (e.g., robotics via Nvidia 2025b), urging certified defenses for world models. By focusing on fine-tuning—a common practice in continual learning—it exposes a "fundamental vulnerability" with actionable insights.

- The paper is well-written and easy to follow.

**Weaknesses:**

- SWAAP assumes that the attacker has full knowledge of the world model architecture (white-box). However, in real-world deployments (e.g., foundational world models like OpenAI 2025 or DeepMind 2024), model details may be black-box or partially black-box. This limits the practical feasibility of the attack, and the paper does not discuss gray-box or black-box variants.

- The first stage of the bi-level optimization employs first-order dynamic barrier gradient descent (Eq. 4), which avoids computing the Hessian but requires $N=16$ mini-updates and extensive buffer sampling (Dall). This makes the approach computationally intensive in high-dimensional continuous environments. The paper does not quantify actual runtime, raising concerns about its applicability to real-time attack scenarios.

- The theorem relies on the gradient of the log-likelihood of continuous transitions, but this assumption may break down under discrete actions or noisy transitions. Furthermore, the paper does not provide guarantees for generalization in non-stationary MDPs with evolving dynamics.

- The approach relies on the dynamic barrier method (Liu et al., 2022), but Fig. 7 only illustrates convergence without proving global optimality or providing bounds under the non-convex objective $J(P_\psi, \pi_\theta)$. Moreover, the constraint $q(\psi, \theta) \leq 0$ may lead to suboptimal target models.

- The experiments are limited to small datasets, where the method shows strong performance under low rp. However, in large-scale fine-tuning scenarios (e.g., millions of trajectories), a higher rp may be required, making the approach easier to detect. The paper does not explore the trade-offs when rp exceeds 20%.

- The method aligns $GrealG_{\text{real}}$​ and $GtargetG_{\text{target}}$​ using cosine dissimilarity, but in high-dimensional embeddings this can lead to gradient explosion or vanishing. The paper does not discuss the sensitivity of the balancing parameter $\alpha$. Thus, how about the impact of $\alpha$?

**Questions:**

Please refer to weaknesses.

---

> ### Author Response · Authors · 2025-11-22
> **Rebuttal (1/3)**
>
> Dear reviewer K1NY:
>
> We sincerely thank the reviewer for the thoughtful and detailed assessment of our work. We appreciate your recognition that SWAAP introduces the first scalable and stealthy data-poisoning attack on world models, and that our bilevel formulation and gradient-matching attack are principled, practical, and clearly presented. We respond to each concern below.
>
> Q1: White-box assumption limits practical feasibility.
>
> A1: We agree with the reviewer that assuming full white-box access may not always hold in real-world deployments. Our goal in this initial work was to establish a strong attack against world models under the most informative setting.
>
> To assess SWAAP’s robustness under weaker attacker capabilities, we conducted a preliminary gray-box experiment in which the attacker only has access to a surrogate world model trained on a single task, while the victim employs a multi-task world model. The attacker performs Stage 1 and Stage 2 entirely on the surrogate model and then supplies the resulting poisoned dataset to the victim for fine-tuning.
>
> In this experiment, the surrogate is a world model trained on MW-Soccer, whereas the victim is a multi-task world model used across multiple tasks. We use $r_p = 0.1, \alpha = 0.1$ in this experiment.  Our initial results show that SWAAP still induces substantial performance degradation across multiple downstream tasks (see table below), despite the mismatch between the surrogate and victim models.
>
> | Source Task | Target Task | Setting  | Return (± std) | δ (± std)     |
> |-------------|-------------|----------|------------------|----------------|
> | MW-Soccer   | MW-Soccer   | Natural  | 1380 ± 382       | 0.11 ± 0.11    |
> | MW-Soccer   | MW-Soccer   | SWAAP    | 1115 ± 442       | 0.13 ± 0.12    |
> | MW-Soccer   | MW-Push     | Natural  | 1290 ± 609       | 0.13 ± 0.16    |
> | MW-Soccer   | MW-Push     | SWAAP    | 1064 ± 692       | 0.16 ± 0.19    |
>
> Q2: Computational cost is not reported.
>
> A2: We agree that reporting resource usage helps readers assess the practicality of SWAAP. Our current measurements are as follows. Stage 1 takes approximately 1 hour on DMControl and Metaworld and 2 hours on Myosuite. Stage 2 takes roughly 10 minutes. To the best of our knowledge, SWAAP is the first framework to study poisoning attacks on world models in MBRL, so there is no existing method that directly matches our full two-stage pipeline. For context, however, prior state-perturbation attacks in RL such as SA-RL [1] and PA-AD [2] require approximately 1.5 hours to train an attacker using an RL algorithm on comparable DMControl tasks. This shows that our Stage 1 compute cost is on the same order as prior attack methods, while Stage 2 remains highly efficient.
>
> Furthermore, our attack designs intentionally reduce computational overhead:
> (1) Stage 1 uses a **first-order dynamic barrier method**, avoiding the expensive Hessian computations typically required in bilevel optimization.
> (2) Stage 2 employs a **modified gradient-matching step** that selects the most influential data points for poisoning instead of exhaustively searching over all possible combinations, which significantly reduces computational cost.
>
> [1] Zhang, H. et al., Robust Deep Reinforcement Learning against Adversarial Perturbations on State Observations. NeurIPS, 2020.
>
> [2] Sun, Y. et al., Who Is the Strongest Enemy? Towards Optimal and Efficient Evasion Attacks in Deep RL. ICLR, 2022.

---

> ### Author Response · Authors · 2025-11-22
> **Rebuttal (2/3)**
>
> Q3:  Assumptions of continuous transitions and limitations under discrete/noisy settings/nonstationary MDP.
>
> A3: Our algorithm is applicable to both discrete and continuous state and action spaces. In particular, the policy gradient applies to both discrete and continuous settings, and our transition gradient theorem (Theorem 1) is derived for transitions with discrete state-action pairs, although the same approach generalizes to continuous settings.
>
> We agree that realistic dynamic models include noise from various sources, including environmental and measurement noise, which affect both model training and online decision-making. In particular, noises in training data make the pre-trained model inaccurate. For example, we observed that even the clean TD-MPC2 dynamics model deviates from the true environment dynamics by around 5 to 10 percent. Our approach leverages this fact to identify a manipulated world model that is hard to distinguish from the clean one. On the other hand, noises in observations at the inference time complicate the agent’s decisions even without attacks, which can potentially be exploited by an advanced attacker. This is orthogonal to our current work and is an interesting direction for future exploration.
>
> While we agree that it is important to consider model-based agents in evolving environments that require continual learning and repeated fine-tuning, our goal is not to solve the broader challenge of generalization in non-stationary MDPs. Model-based RL in non-stationary settings is a difficult and largely independent problem in its own right. As the first work on world-model poisoning, we focus on the stationary case, where the environments used for training and testing remain fixed.
>
> Q4: Dynamic-barrier method does not guarantee global optimality
>
> A4: Our bilevel optimization framework builds on the dynamic barrier framework in [3], where it is shown that the first-order approach converges non-asymptotically to local stationary points even for non-convex objectives. Similar convergence behavior holds in our setting. However, establishing global optimality for bilevel optimization with non-convex objectives remains a difficult, well-known open problem and is not the focus of this work. Our goal in Stage 1 is not to find the globally optimal solution, but rather an effective adversarial and stealth-constrained target model that can be realized through data poisoning. Figure 7 in the Appendix is intended to illustrate the stable convergence of this local optimization process. We will clarify that SWAAP is designed to identify a realizable adversarial model, one that Stage 2 can reliably approximate, rather than a theoretical global optimum.
>
> [3] Ye et al., Bome! bilevel optimization made easy: A simple first-order approach. NeurIPS 2022.
>
> Q5: Large finetune datasets results and different $r_p$ results.
>
> A5: Due to computational resource limitations, we were unable to experiment with million-scale fine-tuning buffers. However, we have conducted additional experiments with fine-tuning sizes ranging from 5,000 to 25,000 with $\alpha = 0.9$ and $r_p = 0.1$ in HumanoidWalk in the following.
>
> | Fine-tuning Dataset Size | Return (R ± std) | δ (delta ± std) |
> |----------------------|------------------|------------------|
> | 5,000                | 521 ± 332        | 0.14 ± 0.08      |
> | 10,000               | 540 ± 348        | 0.14 ± 0.08      |
> | 15,000               | 550 ± 306        | 0.14 ± 0.08      |
> | 20,000               | 607 ± 300        | 0.13 ± 0.08      |
> | 25,000               | 679 ± 234        | 0.12 ± 0.07      |
>
> Although increasing the fine-tuning dataset size reduces the magnitude of performance degradation since more clean data naturally dilutes the poisoned portion, SWAAP still consistently induces meaningful return drops across all tested buffer sizes, demonstrating its robustness even when the victim uses substantially larger fine-tuning datasets.
>
> We also provide an extended ablation on various poisoning ratios $r_p$. The following table reports results for Humanoid-Walk with $\alpha = 0.9$:
>
> | $r_p$  | Return (± std)      | δ (± std)     |
> |------|------------------------|----------------------|
> | clean | 866 ± 57              | 0.09 ± 0.05          |
> | 0.01 | 844 ± 94              | 0.10 ± 0.05          |
> | 0.05 | 739 ± 172             | 0.11 ± 0.07          |
> | 0.10 | 521 ± 332             | 0.14 ± 0.08          |
> | 0.20 | 317 ± 339             | 0.18 ± 08            |
> | 0.30 | 242 ± 305             | 0.19 ± 0.08          |
>
> This ablation clearly shows the trade-off between attack strength and stealthiness, and demonstrates that SWAAP remains effective even at 1%-5% poisoning, while larger $r_p$ naturally increases detectability. More results on this can be found in Appendix G.1, Figure 5.

---

> > ### Author Response · Authors · 2025-11-22
> > **Rebuttal (3/3)**
> >
> > Q6: Cosine dissimilarity in high-dimensional embeddings may cause gradient explosion or vanishing.
> >
> > A6: We thank the reviewer for raising this concern. Our use of cosine similarity directly follows Geiping et al. [4], where the method was used for poisoning high-dimensional images, a regime with far higher dimensionality than our state-transition embeddings.
> >
> > Empirically, across all tested $\alpha$ and $r_p$​, we did not observe gradient explosion or vanishing. Optimization remained numerically stable throughout Stage 2. We will state this empirical observation explicitly in the Appendix of our revision.
> >
> > [4] Geiping et al., Witches' Brew: Industrial Scale Data Poisoning via Gradient Matching. ICLR 2021.
> >
> > Q7: Sensitivity of $\alpha$
> >
> > A7: We include an ablation on $\alpha \in \{0.1, 0.5, 0.9\}$ across several environments in Appendix G.1, Figure 5. A higher $\alpha$ constrains model deviation more strictly, reducing attack strength but improving stealth. A lower $\alpha$ relaxes the deviation constraint and increases performance degradation. We will provide a clearer summary of these findings in Appendix G.
> >
> > We thank the reviewer again for their constructive comments. We will incorporate the requested clarifications, runtime reporting, gray-box discussion, and additional ablations into the final version.

---

> > > ### Comment · Reviewer_K1NY · 2025-11-28
> > >
> > > Thank you for your responses! The responses address most of my concerns. However, I still have a few remaining issues.
> > >
> > > 1. Real-world scenarios are crucial for a security paper, and I believe this work should also explore black-box settings.
> > >
> > > 2. For Q6, no observation in the empirical study does not mean that the issues do not exist. The author should face up to this issue.
> > >
> > > Thus, I prefer to keep my score.

---

> > > > ### Author Response · Authors · 2025-11-28
> > > >
> > > > Dear Reviewer K1NY:
> > > >
> > > >
> > > > We are glad that we have addressed most of your concerns, and thank you for sharing your remaining concerns. We address them below.
> > > >
> > > > Q1: Real-world scenarios are crucial for a security paper, and I believe this work should also explore black-box settings.
> > > >
> > > > A1: We respectfully disagree that real-world world-model systems are exclusively black-box. Several widely used world models are fully open-sourced and publicly documented, such as V-JEPA [1] and Hunyuan3D [2] , making white-box access realistic in many cases. This is also highlighted in a recent survey on the risks of open-weight models [3]. As the first work on world-model data poisoning, we followed common practice in adversarial machine learning research by starting with a white-box setting.
> > > >
> > > > That said, we appreciate the reviewer’s suggestion and have conducted an additional gray-box experiment where the attacker does not know the victim model architecture or task and only has access to a clean environment and a surrogate model. The results in the earlier responses show that SWAAP remains effective even without architectural knowledge. We also note that it is common to allow black-box attacks to interact with a clean environment [4] or train a surrogate model by the attacker [5].
> > > >
> > > > We emphasize that world-model data poisoning differs fundamentally from adversarial RL attacks, and the community currently lacks a formal definition of black-box threat models in this domain. As the first paper in this area, we believe starting with a white-box setting is reasonable and follows the historical progression seen in adversarial supervised learning and RL research.
> > > >
> > > > Q2: For Q6, no observation in the empirical study does not mean that the issues do not exist. The author should face up to this issue.
> > > >
> > > > A2: We thank the reviewer for raising this point. We respectfully argue that cosine similarity itself does not inherently cause gradient explosion or vanishing. Whether such issues arise depends on multiple factors such as network depth, activation functions, and the distribution of latent representations. Prior work has examined this phenomenon in detail. For example, [6] shows that vanishing gradients can occur under two specific geometric conditions of the feature embeddings, and also demonstrates that gradient normalization and cut-initialization can effectively mitigate these issues.
> > > >
> > > > Furthermore, cosine-based gradient matching has been widely adopted in the data poisoning literature [7][8], including settings with significantly higher dimensionality than ours. None of these works report exploding or vanishing gradients in their work.
> > > >
> > > > While analyzing the full theoretical landscape of cosine-similarity optimization dynamics is an interesting direction, it is orthogonal to the core contribution of our work and falls outside our scope.
> > > >
> > > >
> > > > [1] V-JEPA. Meta AI, 2024. https://github.com/facebookresearch/jepa
> > > >
> > > > [2] Hunyuan3D. Tencent AI Lab, 2024. https://github.com/Tencent-Hunyuan/Hunyuan3D-2
> > > >
> > > > [3] Casper et al., Open Technical Problems in Open-Weight AI Model Risk Management. 2025
> > > >
> > > > [4] Gleave et al., Adversarial Policies: Attacking Deep Reinforcement Learning. ICLR 2020.
> > > >
> > > > [5] Huang et al., SEBA: Sample-Efficient Black-Box Attacks on Visual Reinforcement Learning. Arxiv 2025.
> > > >
> > > > [6] Draganov et al., The Hidden Pitfalls of the Cosine Similarity Loss, HiLD 2024.
> > > >
> > > > [7] Geiping et al.,  Witches’ Brew: Industrial Scale Data Poisoning via Gradient Matching, ICLR 2021.
> > > >
> > > > [8] Zhao et al., Dataset Condensation with Gradient Matching, ICLR 2021.

---

### Official Review · Reviewer_5XAj · 2025-10-29

**Soundness:** 3
**Presentation:** 3
**Contribution:** 3
**Rating:** 6
**Confidence:** 5

**Summary:**

The paper presents the first, to my knowledge, study of poisoning attacks against model based reinforcement learning (MBRL). The attacker's objective in the authors' formulation is to minimize the agent's test time return while producing a world model that is less detectable, according to some reasonable metrics like residuals.

The attack has two phases. In the first phase the attacker tries to find a malicious world model $\hat{\psi}$ such that maximizing return under the dynamics of $\hat{\psi}$ also minimizes return in the true task. To achieve this the authors perform some fairly intricate derivations make their bilevel optimization objective more computationally viable. In the second phase the attacker poisons the fine-tuning dataset of the agent's world model such that they learn dynamics close to those of $\hat{\psi}$. In both phases regularization terms are used to minimize the distance between $\hat{\psi}$ and the true dynamics along with reducing the number of poisoned samples.

The attack is evaluated over 9 continuous control tasks from 3 libraries where it achieves respectable results. The detectability of the attack by defenses is also evaluated, showing that some naive defense techniques may not work.

**Strengths:**

* As said in the summary, this is the first paper I'm aware of that studies poisoning attacks in MBRL.

* The proposed method seems to come from reasonable foundational principals -- starting from a reasonable but computationally complex objective and then iteratively approximating or reducing the problem to be more feasible.

* The authors make decent attempts to study and consider defense techniques against their method which is always appreciated.

* The method is designed in a way to not fundamentally violate the dynamics of the true environment, making the attack objective harder but more realistic.

* Given this is the first paper studying this problem, the author's chosen baselines are reasonable and their results are respectable.

**Weaknesses:**

### Attack Objective

In general I find the focus of RL poisoning attack papers on universal availability attacks to be a bit questionable. Here the objective of the attack is to simply minimize the agent's return across all states (subject to some reasonable constraints). As previously stated, this means SWAAP is an availability attack (minimizing utility) and is also universal (applies to all inputs).

The problem with this objective formulation, in my opinion, is that it has a fundamental detectability problem that this paper doesn't really address directly. Let's imagine the "perfect attack" in this setting: the residuals of the poisoned data are indistinguishable from those of the clean data and yet the agent's return is completely minimized, receiving the lowest possible return. In this case the practitioners designing and training the system will for sure not deploy the agent and will try to investigate the issue. In my opinion the same holds for results like those presented in this paper. A 32% performance decrease, on pen-twirl-hand for instance, will certainly be visibly noticable by practitioners as a failed training run. Real deployed systems, e.g. robotic agents, undergo rigorous testing before they're deployed to ensure safety and performance stability. Therefore an objective of universally minimizing the agent's return seems unreasonable to me.

The objective also doesn't really tell me anything new or unique about MBRL. Universal availability attacks have been known and understood in ML in general for a long time, so it is unsurprising that they also work here.

That being said, this universal objective is (unfortnuately) common within the poisoning attack space in RL so I cannot peanilize the authors too heavily. Though I would have much prefered seeing a more targeted objective.

### Minor Weaknesses

* Figure 1 is a bit hard to follow as currently presented. It would be better if "stage 1" appeared at the top left of the figure instead of towards the middle. Currently it takes some effort to figure out the exact flow of the diagram or where to start in my opinion.

* The threat model is still white box and therefore fairly unrealistic. However, since this is the first paper in this area, the threat model is excusable.

**Questions:**

* What is the authors' motivation or rationale for chosing an adversarial objective of minimizing return?

---

> ### Author Response · Authors · 2025-11-22
> **Rebuttal (1/2)**
>
> Dear Reviewer 5XAj:
>
> We sincerely thank the reviewer for the positive and accurate understanding of our paper. We appreciate your recognition that this is the first work to study poisoning attacks on world models in MBRL, that our bilevel approximation and gradient-matching construction follow sound principles, and that our evaluation and defense analysis demonstrate meaningful and realistic attack behavior. We respond to your concerns point by point below.
>
>
> Q1: Motivation for minimizing return. Could we change our attack objective?
>
> A1: Our rationale aligns with the broader literature on robustness and poisoning in RL: return is the canonical performance metric for RL agents, and minimizing it under constrained deviation provides a clear, measurable notion of vulnerability.
>
> Importantly, due to the modular design of SWAAP, our attack objective can indeed be replaced by other targeted or behavior-specific objectives simply by modifying the Stage-1 formulation. For example, we can adopt a **target-policy attack** objective of the form
>
>
> $min_{ψ, θ}   \textrm{ KL}(θ, θ^†) + λ \textrm{ KL}(P_ψ , P)$
>
> subject to   $θ ∈ \textrm{argmax}_{θ'} J(P_ψ, θ')$.
>
> This allows the attacker to steer the learned policy toward a prescribed target policy $θ^†$. The KL terms may also be replaced with occupancy-based divergences.
>
> A **backdoor-style** targeted policy can also be expressed as:
>
> $π^†(a|s) = π(a|s)$                for normal states $s$
>
> $π^†(a|s̃) = 1[a = a_{bad}]$           for triggered states $s̃$
>
> Thus, targeted attacks require modifying only Stage 1; Stage 2 remains unchanged.
>
> Q2: The objective does not have anything unique about MBRL. Universal availability attacks have been known and understood in ML in general for a long time.
>
> A2: We thank the reviewer for raising this point. While minimizing return is indeed a standard adversarial objective in RL, our contribution lies not in redefining the objective, but in showing that this objective becomes fundamentally different and more challenging, when attacking world models rather than policies or value functions. Unlike classical reward and policy-poisoning attacks[1][2], MBRL has an additional optimization layer, which is policy planning under the learned world model. Moreover, our formulation highlights that world models are vulnerable even when the attacker is constrained to remain close to the true dynamics, a property that cannot be revealed through classical availability attacks on policies.
>
> [1] Zhang et al., Adaptive Reward-Poisoning Attacks against Reinforcement Learning. ICML 2020.
>
> [2] Ma et al., Policy Poisoning in Batch Reinforcement Learning and Control. NeurIPS 2019.
>
> Q3: The victim might notice the reward drop and realize the existence of the attacker.
>
> A3: We thank the reviewer for raising this important point regarding detectability from the victim’s perspective. However, we argue that in a world-model-based RL pipeline, the victim does not have access to the ground-truth transition function, and therefore cannot distinguish whether a lower return comes from suboptimal learning, environment randomness, or an adversarially poisoned model. In practice, world-model performance often fluctuates due to distribution shift, model mis-specification, or imperfect finetuning, making small or moderate return drops difficult to attribute to an attack.
>
> Moreover, SWAAP does not require a large return drop to succeed. For instance, in Humanoid-Walk, our attack induces approximately a 10% reduction in return, which is well within the range of fluctuations commonly observed in model-based RL.
> Finally, Stage 1 of SWAAP can be adjusted to only perturb a certain fraction of the state-action pairs’ transitions. We have conducted additional experiments on perturbing top 10% and 50% most influential state-action pairs measured by $max_{a \in A}Q(s,a) - min_{a' \in A}Q(s,a’)$, compared to perturbing all pairs as reported in our paper in Humanoidwalk with $r_p = 0.1, \alpha = 0.9$.
>
> | Stage 1 poison fraction | Return (± std) | δ (± std) |
> |--------------|-------------------|------------------|
> | 0% (clean)   | 866 ± 57          | 0.09 ± 0.05      |
> | 10%          | 825 ± 106         | 0.10 ± 0.06      |
> | 50%          | 669 ± 242         | 0.12 ± 0.07      |
> | 100%         | 521 ± 332         | 0.14 ± 0.08      |

---

> > ### Author Response · Authors · 2025-11-22
> > **Rebuttal (2/2)**
> >
> > Stage 2 of SWAAP can also adjust how many data points are poisoned and thus directly trade off attack strength and stealth. We conducted an additional experiment on different poisoning ratios $r_p$ with $\alpha = 0.9$ in the Humanoid-Walk environment:
> >
> > | r_p  | Return (± std)      | (delta ± std)     |
> > |------|------------------------|----------------------|
> > | clean | 866 ± 57              | 0.09 ± 0.05          |
> > | 0.01 | 844 ± 94              | 0.10 ± 0.05          |
> > | 0.05 | 739 ± 172             | 0.11 ± 0.07          |
> > | 0.10 | 521 ± 332             | 0.14 ± 0.08          |
> > | 0.20 | 317 ± 339             | 0.18 ± 0.08          |
> > | 0.30 | 242 ± 305             | 0.19 ± 0.08          |
> >
> > These results show that by decreasing the poison rates in both Stage 1 and Stage 2, SWAAP achieves a tunable balance between attack impact and stealthiness, allowing attackers to induce small or moderate return shifts that are within the range of typical world-model deviation.
> >
> > Q4: Figure 1 needs improvement.
> >
> > A4: We thank the reviewer for the suggestion. We have reorganized the figure so that **Stage 1** appears at the top left and the flow of the diagram is clearer, making the pipeline easier to follow.
> >
> > Q5: White-box assumption may be unrealistic.
> >
> > A5:: We agree that the full white-box assumption is strong. To assess SWAAP’s robustness under weaker attacker capabilities, we conducted a preliminary gray-box experiment in which the attacker only has access to a surrogate world model trained on a single task, while the victim employs a multi-task world model. The attacker performs Stage 1 and Stage 2 entirely on the surrogate model and then supplies the resulting poisoned dataset to the victim for fine-tuning.
> > In this experiment, the surrogate is a world model trained on MW-Soccer, whereas the victim is a multi-task world model used across multiple tasks. We use $r_p = 0.1, \alpha = 0.1$ in this experiment. Our initial results show that SWAAP still induces substantial performance degradation across multiple downstream tasks (see the table below), despite the mismatch in between the surrogate and victim models.
> >
> > | Source Task | Target Task | Setting  | Return (± std) | δ (± std)     |
> > |-------------|-------------|----------|------------------|----------------|
> > | MW-Soccer   | MW-Soccer   | Natural  | 1380 ± 382       | 0.11 ± 0.11    |
> > | MW-Soccer   | MW-Soccer   | SWAAP    | 1115 ± 442       | 0.13 ± 0.12    |
> > | MW-Soccer   | MW-Push     | Natural  | 1290 ± 609       | 0.13 ± 0.16    |
> > | MW-Soccer   | MW-Push     | SWAAP    | 1064 ± 692       | 0.16 ± 0.19    |
> >
> > We again thank the reviewer for the thoughtful feedback and constructive suggestions. We will incorporate the revisions and clarifications to further strengthen the paper.

---

> > > ### Comment · Reviewer_5XAj · 2025-11-24
> > > **Response to the Authors**
> > >
> > > Hello,
> > >
> > > Thank you for your detailed rebuttal, additional experiments, and improvements to the paper. I think I still disagree on the utility of a return minimizing objective in practice, but I agree that the paper achieves its goal in showing the vulnerabilities of MBRL to training time attacks.
> > >
> > > I am in support of the paper being accepted as a poster. I look forward to seeing the responses and opinions of the other reviewers.
> > >
> > > Thank you,
> > > Reviewer 5XAj

---

> > > > ### Author Response · Authors · 2025-11-26
> > > >
> > > > Dear Reviewer 5XAj,
> > > >
> > > > Thank you very much for your supportive follow-up and for recommending our paper for acceptance. We appreciate your thoughtful feedback throughout the process, and we will incorporate the suggested clarifications and improvements in the final version.
> > > >
> > > > Sincerely,
> > > >
> > > > Authors of Submission 14438

---

### Official Review · Reviewer_ruCt · 2025-10-30

**Soundness:** 2
**Presentation:** 3
**Contribution:** 3
**Rating:** 4
**Confidence:** 3

**Summary:**

This paper explores poisoning agent's world models in model-based reinforcement learning. First, the attacker identifies the world model that would lead to the worst-case performance of a learning agent while remaining close to the original environment. In the second stage, the attacker perturbs the model so that it matches the target world model from the first stage. This attack was tested on three common RL benchmarks.

**Strengths:**

1. The paper addresses a relevant problem, by extending existing poisoning attacks against RL-agents to the poisoning of world models.
2.  The assumptions of the attack are clearly described and generally realistic. The method itself is formally well-defined.
3. The inclusion of defense mechanisms is a useful addition.

**Weaknesses:**

1. The assumed poisoning rate of 10% is overly high. This is especially problematic, since there appear to be no ablation studies varying the amount of poisoned samples.
2. The results of the proposed SWAAP method show high variance compared to the other baselines, particularly in the clean setting. This makes it difficult to assess the quality of the poisoning attack.
3. It would have been helpful to have more environments, especially since the Myosuite results are extremely volatile.
4. Given the two stage nature of the algorithm, it would be nice to see how well the individual stages work, i.e. is the adversary actually able to discover a good worst-case model, and is the adversary able to poison the model such that it becomes close to the target model?

**Questions:**

1. How does the attack perform at various levels of poisoning?
2. See W4

---

> ### Author Response · Authors · 2025-11-22
> **Rebuttal (1/2)**
>
> Dear Reviewer ruCt:
>
> We sincerely thank the reviewer for the clear summary and thoughtful assessment of our work. We appreciate your recognition that the problem setting is relevant, the assumptions are realistic, the method is well-defined, and that including defense mechanisms adds value. We address your concerns point by point below.
>
> Q1: Poisoning rate of 10% is overly high; lack of ablation on poisoning amount and poisoning level.
>
> A1: We agree that evaluating varying poisoning rates is important. In the original submission, we use a 10% poisoning rate applied to a relatively small finetuning buffer of 5,000 transitions. This means only a few hundred samples are modified, which we believe is realistic in practice. We also include an ablation study in Appendix G that examines the effect of the poisoning rate $r_p​$ and poisoning strength, characterized by the deviation constraint $\alpha$ (ranging from 0.1 to 0.9).
>
> In addition, we conducted further experiments with $r_p$<10%. Below, we report the results on Humanoidwalk with $\alpha = 0.9$.
> | $r_p$ | Return (± std)      | δ ( ± std)     |
> |------|------------------------|----------------------|
> | clean | 866 ± 57              | 0.09 ± 0.05          |
> | 0.01 | 844 ± 94              | 0.10 ± 0.05          |
> | 0.05 | 739 ± 172             | 0.11 ± 0.07          |
> | 0.10 | 521 ± 332             | 0.14 ± 0.08          |
> | 0.20 | 317 ± 339             | 0.18 ± 0.08          |
> | 0.30 | 242 ± 305             | 0.19 ± 0.08          |
>
> We observe that with lower poisoning rates (1% and 5%), the attack performance decreases as expected, but it still consistently induces noticeable degradation in return.
>
> Q2: High variance in SWAAP results compared to baselines makes interpretation difficult.
>
> A2: Thank you for pointing this out. The higher variance is primarily caused by the reward structure of these environments rather than by our attack method itself. In tasks such as pen-twirl-hard, a successful attack episode drives the agent into a failure trajectory and produces extremely low returns (e.g., around 1,000), whereas an unsuccessful attack episode still behaves similarly to clean execution and achieves high returns (e.g., around 5,000). Even clean policies naturally exhibit noticeable variance in these environments because occasional failures lead to large return drops. Under attack, this effect becomes more pronounced: the mixture of low-reward failure episodes and high-reward normal episodes produces a wider spread of returns. We will clarify this point in the paper.
>
> Q3: Limited number of environments
>
> A3: We agree that additional environments would further strengthen the evaluation. We have done extra evaluation on the following environments.
>
> | Environment               | Natural Return (± std) | SWAAP Return (± std) | Natural δ (± std) | SWAAP δ (± std) |
> |---------------------------|---------------------------|--------------------------|--------------------|------------------|
> | MyoSuite – obj-hold-hard  | -289 ± 2343              | -2671 ± 4241            | 0.10 ± 0.07        | 0.13 ± 0.05      |
> | MyoSuite – pose           | 696 ± 4                  | 665 ± 157               | 0.04 ± 0.04        | 0.08 ± 0.06      |
> | MyoSuite – key-turn       | 1125 ± 196               | 940 ± 375               | 0.19 ± 0.14        | 0.11 ± 0.08      |
> | MetaWorld – coffee-pull   | 1511 ± 25                | 1402 ± 318              | 0.08 ± 0.04        | 0.09 ± 0.06      |
> | MetaWorld – door-open     | 1551 ± 58                | 1310 ± 445              | 0.04 ± 0.02        | 0.06 ± 0.05      |
>
> All experiments use $r_p = 0.1$  and are evaluated over 100 episodes. For Obj-Hold-Hard, Key-Turn and Coffee-Pull, we set $\alpha = 0.9$, while for Pose and Door-Open we use $\alpha = 0.1$. These additional environments further confirm that SWAAP can reliably reduce return while keeping the poisoned world model close to the clean model, consistent with the main paper’s findings.

---

> ### Author Response · Authors · 2025-11-22
> **Rebuttal (2/2)**
>
> Q4: How each individual step works.
>
> A4: We appreciate the reviewer’s interest in understanding the contribution of each stage. The effect of Stage 1 (perturbed model identification) is reflected in Table 1: Direct Model Poisoning, where we directly replace the victim’s model with the target world model. This isolates the Stage 1 objective and demonstrates how the identified target dynamics alone can degrade the agent’s performance.
>
> To show the contribution of Stage 2 (data poisoning), we added an experiment that measures the relative deviation δ (defined in line 377 of the paper) between the target model $P_{\hat{\psi}}$ and the clean model $P_{\psi_0}$ and the deviation between the target model  $P_{\hat{\psi}}$  and the poisoned model $P_{\psi}$. In the Humanoid environment with $\alpha = 0.1$ and $r_p = 0.3$, we observed: $\delta(P_{\hat{\psi}}, P_{\psi_0}) = 0.15$ and $\delta(P_{\hat{\psi}}, P_{\psi}) = 0.11$.
>
> This demonstrates that Stage 2 successfully reduces the distance between the victim’s poisoned model and the target model identified in Stage 1, confirming that gradient-matching poisoning effectively guides the model toward the adversarial target.
>
> We thank the reviewer again for the helpful comments. We will incorporate the requested clarifications and additional experiments to strengthen the final version of the paper.

---

> > ### Comment · Reviewer_ruCt · 2025-11-26
> >
> > Thank you for your response.
> >
> > I believe my concerns W3 and W4 have been adequately addressed. However, I am not convinced by the response to the other two points.
> >
> > W1: I understand that, in absolute terms, the poisoning rate is relatively low. Nevertheless, poisoning 10% of a dataset seems to make detecting an attack quite easy. I believe that experiments with larger datasets would be necessary to evaluate this aspect.
> >
> > W2: I don't see the point of including the variance over the evaluated episodes. It would be more interesting to see the variance over the average return when repeating the attack over multiple distinct seeds. In the current setup, these variance numbers are essentially meaningless.
> >
> > While I like the paper, I don't believe it is ready for publication in its current form. Therefore, I will maintain my current score.

---

> ### Author Response · Authors · 2025-11-29
>
> Dear Reviewer ruCt,
>
> We are glad that we have addressed most of your concerns, and thank you for the follow-up comments. We address your remaining concerns below.
>
> Q1: Detectability when poisoning 10% of the dataset; need larger-dataset results.
>
> A1: We respectfully disagree that a 10% poison rate makes SWAAP easy to detect. As shown in Figure 4(a), the pre-training detection method De-Pois [1] cannot distinguish between poisoned and clean data—residual distributions are statistically indistinguishable. This is precisely due to the regularization term in the Stage 1 objective and the loss design in Stage 2, which explicitly penalize large or conspicuous perturbations.
>
> We agree that evaluating larger fine-tuning sets is valuable. In our previous response, we reported results at smaller poisoning rates ($r_p=0.01, 0.05$) on a 5k buffer:
>
> | $r_p$  | Return (R ± std) | δ (delta ± std) |
> |------|------------------|------------------|
> | clean | 866 ± 57         | 0.09 ± 0.05      |
> | 0.01 | 844 ± 94         | 0.10 ± 0.05      |
> | 0.05 | 739 ± 172        | 0.11 ± 0.07      |
>
> Even at 1% poisoning rate, SWAAP remains effective.
>
> To further address your concern, we ran additional experiments using a *fixed* set of 500 poisoned samples while increasing the fine-tuning dataset sizes to 10k and 15k. The results are below:
>
> | Fine-tuning Dataset Size | Reward (± std)     | δ (± std)     |
> |------------------------|------------------------|----------------------|
> | 5,000                  | 521 ± 332              | 0.14 ± 0.08          |
> | 10,000                 | 634 ± 269              | 0.13 ± 0.07          |
> | 15,000                 | 796 ± 150              | 0.10 ± 0.06          |
>
> This new result shows that our SWAAP attack remains effective with a fixed number of poisoned data samples and a larger fine-tuning dataset.
>
>
> We will incorporate these findings and clarify detectability discussions in the final version.
>
> Q2: Variance should reflect multiple seeds, not variance of multiple episodes.
>
> A2: We appreciate this clarification. It is a common practice in previous RL attack and defense works (e.g., [2][3][4]) to report the variance of multiple episodes' return. To address your concern, we have now conducted experiments on Humanoid-Walk using 10 independent random seeds, each with 100 evaluation episodes, with all other settings identical to those in Table 1 of the main text (fine-tuning data size $= 5,000$, $r_p=0.1$, $\alpha=0.9$). We report the variance across the seed-level average as follows: Return $= 565\pm20$, $\delta = 0.139\pm0.003$. This is consistent with the result presented in the paper (Return $= 521 \pm 332$, $\delta = 0.14\pm0.08$), as the variance here is computed over the mean return of 100 episodes.
>
> We will include these results and update the experimental description to clarify both types of variance reporting.
>
> We thank you again for the constructive feedback. We believe the revised results and clarifications address your remaining concerns, and we kindly ask you to reconsider your rating accordingly.
>
> Sincerely,
> Authors of Submission 14438
>
>
> [1] Chen et al., De-pois: Anattack-agnostic defenseagainstdatapoisoningattacks. IEEE Transactions on Information Forensics and Security, 2021
>
> [2] Zhang et al., Robust Deep Reinforcement Learning against Adversarial Perturbations on State Observations. NeurIPS, 2020.
>
> [3] Sun et al., Who Is the Strongest Enemy? Towards Optimal and Efficient Evasion Attacks in Deep RL. ICLR, 2022.
>
> [4] Sun and Zheng, Belief-Enriched Pessimistic Q-Learning against Adversarial State Perturbations. ICLR, 2024.

---

### Official Review · Reviewer_vpn2 · 2025-11-01

**Soundness:** 3
**Presentation:** 3
**Contribution:** 3
**Rating:** 6
**Confidence:** 4

**Summary:**

This paper proposes SWAAP, a novel data poisoning attack against the world model. SWAAP is a two-phase attack strategy, comprising the identification of a worst-case target world model via bi-level optimization, and the data poisoning process that guides the benign world model toward the target world model through gradient matching. Experiments show SWAAP can effectively degrade the TD-MPC2 agent's performance across various environments and tasks.

**Strengths:**

1. This paper is well written and easy to follow.
2. The theoretical part is solid. Although both the bi-level optimization algorithm (Liu et al., 2022) and the gradient matching algorithm (Geiping et al., 2021) are not novel, the joint application of these two algorithms for data poisoning is noteworthy.
3. Experiments show SWAAP is effective for degrading the RL agent's performance via only poisoning a small subset of the fine-tuning dataset.
4. Various defenses are discussed, and SWAAP can still work even under defense.

**Weaknesses:**

1. The backbone world model algorithm is limited. Comparing different backbone algorithms (e.g., DreamerV3 and DINO-WM, as mentioned by the authors) would benefit this paper.
2. The results in Table 1 and Figure 2 are not consistent. In Table 1, SWAAP obtains 776 in DMControl humanoid-walk, while in Figure 2, SWAAP obtains a score within [400-600] under $\alpha=0.9$. The authors should clearly explain this difference to ensure the reliability of the results.
3. There is no discussion on the computation resources and costs. Since SWAAP is a two-stage complex optimization algorithm, it is beneficial to illustrate the efforts required by the attacker to launch a successful attack, such as the time.

**Questions:**

Please refer to Weaknesses.

---

> ### Author Response · Authors · 2025-11-22
>
> Dear Reviewer vpn2:
>
> We sincerely thank the reviewer for the positive and precise understanding of our paper. We appreciate your recognition that the joint use of dynamic-barrier bilevel optimization and gradient-matching poisoning is technically solid and novel in the MBRL context. We will address your concerns point by point.
>
> Q1: More backbone world-model algorithms.
>
> A1: We agree that evaluating additional backbone world models (e.g., DreamerV3, DINO-WM) would further strengthen the generality of our method. Our initial choice of TD-MPC2 was motivated by its unified world-model and policy optimization pipeline and its widespread adoption in general-purpose agents. While our two-stage attack framework is general, different world-model backbones require non-trivial engineering effort to adapt our bilevel optimization and gradient matching pipeline. We view this as a valuable direction for future work and are excited about extending our framework to a broader set of models.
>
> In this spirit, we have begun running experiments on DINO-WM. Due to the time and computational resource constraints, we were only able to complete Stage 1. Our initial results show that the identified target world model decreases the success rate of the DINO-WM Push-T task from 92% with a relative deviation $\delta = 0.11 \pm 0.06$ to 72% with relative deviation $\delta = 0.24 \pm 0.10$. We are currently implementing Stage 2 and will report the complete results in the revised version.
>
>
> Q2: Inconsistency between Table 1 and Figure 2
>
> A2: Thank you for pointing out this discrepancy. The difference arises because Table 1 uses $\alpha = 0.99$ for the DMControl humanoid-walk experiment (to keep the deviation close to the clean model deviation), while Figure 2 uses $\alpha = 0.9$. The value of $\alpha$ controls the tradeoff between attack performance and the amount of perturbations introduced, and different $\alpha$ values naturally produce different performance levels, even under the same poisoning ratio. We will replace the result in Table 1 with $\alpha = 0.9$ to keep consistency.
>
> Q3: Computation resources and attack cost
>
> A3: We agree that reporting resource usage helps readers assess the practicality of SWAAP. Our current measurements are as follows. Stage 1 takes approximately 1 hour on DMControl and Metaworld and 2 hours on Myosuite. Stage 2 takes roughly 10 minutes. To the best of our knowledge, SWAAP is the first framework to study poisoning attacks on world models in MBRL, so there is no existing method that directly matches our full two-stage pipeline. For context, however, prior state-perturbation attacks in RL such as SA-RL [1] and PA-AD [2] require approximately 1.5 hours to train an attacker using an RL algorithm on comparable DMControl tasks. This shows that our Stage 1 compute cost is on the same order as prior attack methods, while Stage 2 remains highly efficient.
>
> Furthermore, our attack designs intentionally reduce computational overhead:
> (1) Stage 1 uses a **first-order dynamic barrier method**, avoiding the expensive Hessian computations typically required in bilevel optimization.
> (2) Stage 2 employs a **modified gradient-matching step** that selects the most influential data points for poisoning instead of exhaustively searching over all possible combinations, which significantly reduces computational cost.
>
> We thank the reviewer again for the helpful feedback. We will incorporate the clarifications and additional results to strengthen the final version of the paper.
>
> [1] Zhang, H. et al. Robust Deep Reinforcement Learning against Adversarial Perturbations on State Observations. NeurIPS, 2020.
>
> [2] Sun, Y. et al. Who Is the Strongest Enemy? Towards Optimal and Efficient Evasion Attacks in Deep RL. ICLR, 2022.

---

> > ### Author Response · Authors · 2025-12-03
> > **Follow Up Results on DINO-WM**
> >
> > Dear Reviewer vpn2:
> >
> > As promised in our previous response, we conducted further experiments incorporating Stage 2 of our attack framework on DINO-WM. Using the perturbed model identified in Stage 1, we applied gradient matching to craft poisoned samples (with a poisoning ratio of 15%) and fine tuned a DINO world model on the resulting poisoned dataset. Evaluation on the Push-T task shows a performance drop from 92% to 77% with $\delta = 0.16 \pm 0.09$. These preliminary results on DINO-WM and goal-conditioned tasks show that our method can be applied to more complex and diverse settings.
> >
> > Sincerely,
> >
> > Authors of Submission 14438

---

### Author Response · Authors · 2025-12-03
**Rebuttal Summary**

Dear AC,

Thank you for your effort in serving as AC under this year’s unforeseen circumstances. We would like to take this opportunity to summarize our work and officially conclude the discussion phase.

Overall, the reviewers highlighted several strengths of our work, including:
- It is the first study to investigate poisoning attacks in model-based RL.
- The joint use of dynamic-barrier bilevel optimization and gradient matching for data poisoning is technically solid and novel.
- The paper makes decent attempts to analyze and discuss potential defense techniques.
- The paper is well written and easy to follow.

Below is a concise summary of how we addressed reviewers’ concerns during the rebuttal period:

- **Reviewer vpn2**
  - Added a preliminary experiment on DINO-WM to address the concern about limited backbone world models.
  - Reported detailed computational cost to address the missing runtime analysis.

- **Reviewer ruCt**
  - Added new results for multiple poison rates $r_p$ and expanded ablation studies to show the contribution of each stage.
  - Reported results under larger fine-tuning dataset sizes (5k-15k).
  - Provided mean ± std results on average return and deviation over 10 independent seeds to address concerns about variance.

- **Reviewer 5XAj**
  - Provided a targeted-policy attack formulation to show that our framework naturally extends beyond return minimization.
  - Provided results on only changing a certain fraction of state-action pairs in Stage 1 to address the concern that our SWAAP attack must attack all state-action pairs.
  - Added a preliminary gray-box experiment to address concerns about white-box limitations.

- **Reviewer K1NY**
  - Explained concerns regarding global optimality, applicability to discrete/noisy transitions, and limitations under non-stationary MDPs. We also want to mention that these aspects are orthogonal to the core contribution of our work.
  - Added additional experiments with varying $r_p$, different fine-tuning dataset sizes, different $\alpha$ levels, and gray-box settings.
  - Provided an analysis explaining why white-box/gray-box settings remain realistic in practical world-model deployments.
  - Addressed concerns about cosine similarity by explaining empirical stability and referencing existing analyses on its gradient behavior.

- **Reviewer 5XAj** additionally expressed support for accepting the paper, noting that the work achieves its goal of demonstrating vulnerabilities of MBRL.

We believe we have thoughtfully and substantially addressed all concerns raised by the reviewers, and we sincerely hope that our rebuttal will be taken into consideration during the final decision process.

Authors of Submission 14438

---

### Meta-Review · Area_Chair_K5i4 · 2026-01-03

**Summary:**

The submission proposes a data poisoning attack against the world model, named SWAAP. SWAAP is a two-phase attack strategy, comprising the identification of a worst-case target world model via bi-level optimization, and the data poisoning process that guides the benign world model toward the target world model through gradient matching.

As pointed by Reviewer vpn2, the author should have revised the wording in the paper. From reviewer vpn2: The bi-level optimization algorithm was proposed by Liu et al., 2022 and the gradient matching algorithm by Geiping et al., 2021, thus, only the joint application of these two algorithms to data poisoning is novel. Reviewer ruCt is still concerned with the detectability of the attack. The black vs gray vs white box access has not been fully resolved with reviewer K1NY. Finally, reviewer 5XAj disagrees on the utility of a return minimizing objective in practice.

Overall, while some reviewers explicitly state that the paper should not be accepted in the current form (reviewer ruCt), other reviewers recommend only the acceptance as a poster (reviewer 5XAj). There is no clear consensus among reviewers regarding this work and not a clean supporter of this submission. Given the current state of the work, I recommend rejection.

**Reviewer Concerns:**

As pointed by Reviewer vpn2, the author should have revised the wording in the paper. From reviewer vpn2: The bi-level optimization algorithm was proposed by Liu et al., 2022 and the gradient matching algorithm by Geiping et al., 2021, thus, only the joint application of these two algorithms to data poisoning is novel.

Reviewer ruCt is still concerned with the detectability of the attack. The authors added more environments, as requeste by the reviewer.

Reviewer 5XAj disagrees on the utility of a return minimizing objective in practice, but agrees that the paper achieves its goal in showing the vulnerabilities of MBRL to training time attacks.

Reviewer K1NY requested the experiments in the black-box setting but only preliminary experiments in the gray-box setting were provided. The authors argue in the last comment that the white-box access is still realistic in many scenarios. The authors reported required computational overhead and provided an ablation on various poisoning ratios.

**Reviewer Scores:**

vpn2: 6

ruCt: 4

5XAj: 6

K1NY: 4

---

### Decision · Program_Chairs · 2026-01-26

Reject